

**Seasonal Carbonate Chemistry Variability in Marine Surface Waters**
**of the Pacific Northwest**
Andrea J. Fassbender[1,2,*], Simone R. Alin[2], Richard A. Feely[2], Adrienne J. Sutton[2], Jan A.
Newton[3], Christopher Krembs[4], Julia Bos[4], Mya Keyzers[4], Allan Devol[5], Wendi Ruef[5], and Greg
Pelletier[4]
[1]Monterey Bay Aquarium Research Institute, Moss Landing, CA 95039, USA
[2]Pacific Marine Environmental Laboratory, National Oceanic and Atmospheric Administration,
Seattle, WA 98115, USA
[3]Applied Physics Laboratory, University of Washington, Seattle, WA 98105, USA
[4]Washington State Department of Ecology, Olympia, WA 98504, USA
[5]School of Oceanography, University of Washington, Seattle, WA 98195, USA
*Corresponding author: fassbender@mbari.org
**Abstract**
Fingerprinting ocean acidification (OA) in U.S. West Coast waters is extremely challenging
due to the large magnitude of natural carbonate chemistry variations common to these regions.
Additionally, quantifying a change requires information about the initial conditions, which is not
readily available in most coastal systems. In an effort to address this issue, we have collated high-
quality, publicly-available data to characterize the modern seasonal carbonate chemistry variability
in marine surface waters of the Pacific Northwest. Underway ship data from Version 4 of the
Surface Ocean $CO_2$ Atlas, discrete observations from various sampling platforms, and sustained
measurements from regional moorings were incorporated to provide ~100,000 inorganic carbon
observations from which modern seasonal cycles were estimated. Underway ship and discrete
observations were merged and gridded to a 0.1°×0.1° scale. Eight unique regions were identified
and seasonal cycles from grid cells within each region were averaged. Data from nine surface
moorings were also compiled and used to develop robust estimates of mean seasonal cycles for
comparison with the eight regions. This manuscript describes our methodology and the resulting
mean seasonal cycles for multiple OA metrics in an effort to provide large-scale, environmental
context for ongoing research, adaptation, and management efforts throughout the Pacific





Northwest. Major findings include the identification of unique chemical characteristics across the study domain. There is a clear increase in the ratio of dissolved inorganic carbon (DIC) to total alkalinity (TA) and in the seasonal cycle amplitude of carbonate system parameters when moving from the open ocean North Pacific into the Salish Sea. Due to the logarithmic nature of the pH scale (pH = $-\log_{10}[H^+]$, where $[H^+]$ is the hydrogen ion concentration), lower annual mean pH values (associated with elevated DIC:TA) coupled with larger magnitude seasonal pH cycles results in seasonal $[H^+]$ ranges that are ~27 times larger in Hood Canal than in the neighboring North Pacific open ocean. Organisms living in the Salish Sea are thus exposed to much larger seasonal acidity changes than those living in nearby open ocean waters. Additionally, our findings suggest that lower buffering capacities in the Salish Sea make these waters less efficient at absorbing anthropogenic carbon than open ocean waters at the same latitude.

All data used in this analysis are publically available at the following websites:
- Surface Ocean $CO_2$ Atlas Version 4 coastal data, https://doi.pangaea.de/10.1594/PANGAEA.866856;
- National Oceanic and Atmospheric Administration (NOAA) West Coast Ocean Acidification cruise data, 10.3334/CDIAC/otg.CLIVAR_NACP_West_Coast_Cruise_2007; 10.3334/CDIAC/OTG.COAST_WCOA2011; 10.3334/CDIAC/OTG.COAST_WCOA2012; 10.7289/V5C53HXP;
- University of Washington (UW) and Washington Ocean Acidification Center cruise data, https://doi.org/10.5281/zenodo.1184657;
- Washington State Department of Ecology seaplane data, https://doi.org/10.5281/zenodo.1184657;
- NOAA Moored Autonomous $p$CO_2 (MAPCO2) Buoy data, https://doi.org/10.3334/CDIAC/OTG.TSM_LAPUSH_125W_48N; https://doi.org/10.3334/CDIAC/OTG.TSM_WA_125W_47N; https://doi.org/10.3334/CDIAC/OTG.TSM_DABOB_122W_478N; https://doi.org/10.3334/CDIAC/OTG.TSM_TWANOH_123W_47N;
- UW Oceanic Remote Chemical/Optical Analyzer Buoy data, https://doi.org/10.5281/zenodo.1184657;
- NOAA Pacific Coast Ocean Observing System cruise data, https://doi.org/10.5281/zenodo.1184657.

## 1. Introduction

Changes in seawater chemistry caused by the continuous absorption of anthropogenic carbon dioxide ($CO_2$) from the atmosphere are altering the marine environment in ways that are often invisible to the human eye but may resonate throughout coastal communities in the form of altered



ecosystem health and economic vulnerabilities associated with food security (e.g., Adelsman and
Binder, 2012; Barton et al., 2015; Ekstrom et al., 2015; Somero et al., 2016; Wong et al., 2014).
Rising $CO_2$ partial pressures ($pCO_2$) and hydrogen ion concentrations ($[H^+]$), commonly referred
to as "Ocean Acidification" (OA; Caldeira and Wickett, 2003; Doney et al., 2009), have been
observed throughout the global open ocean (e.g., Bates et al., 2014; Brewer, 1978; Feely et al.,
2004, 2009; Sabine, 2004). Yet, in the coastal zone it remains challenging to attribute carbonate
chemistry changes to anthropogenic factors due to the much larger and more sporadic natural
chemical variations that take place near shore (Borges, 2011; Evans et al., 2011; Fassbender et al.,
2016; Feely et al., 2016; Harris, 2013; Takeshita et al., 2015; Waldbusser and Salisbury, 2014).
Unlike the open ocean, the coastal zone is complicated by land-sea interactions (e.g., river input,
upwelling, point source pollution, as well as tidal and estuarine mixing) that influence water and
chemical residence times as well as carbon transformation processes, and these may also be
changing over time due to human activities (Bauer et al., 2013; Cai et al., 2011; Feely et al., 2016;
Raymond et al., 2008; Raymond and Cole, 2003; Regnier et al., 2013). Baseline observations
capturing the range of modern variability in the coastal ocean will therefore be required to
accurately fingerprint the ongoing carbonate system changes associated with human perturbations
to the global climate system.

Washington State has an expansive coastline where a diversity of land-sea interfaces results in
regional patchiness, requiring local assessments of OA that cannot be inferred from the broader
North Pacific Ocean (e.g., Feely et al., 2010). Monthly pH observations have been collected
throughout Washington waters since 1989 by the Washington State Department of Ecology (ECY)
using electrode-based pH measurements with known accuracy and precision limitations. The use
of a consistent method has allowed ECY to perform intensive quality assurance (Bos et al., 2016),
leading to the detection of interannual variability and secular trends in pH over the data record.
Still, great uncertainty remains in how other carbonate system parameters have changed
throughout the region. In 2006, the National Oceanic and Atmospheric Administration (NOAA)
and the University of Washington (UW) began sustained high-frequency (3-hour) observations of
carbonate chemistry in Washington waters with the deployment of an autonomous $CO_2$ sensor on
a surface mooring. Three additional surface moorings have been outfitted with $CO_2$ sensors since
2009 (Alin et al., 2015). Routine discrete sampling of two carbonate system parameters conducted



by NOAA and regional partners started in 2007 (Feely et al., 2008, 2010, 2016), and the
deployment of autonomous pH sensors on regional surface moorings was initiated by the UW and
NOAA in 2010. In addition to these targeted carbon observations, $CO_2$ measurements collected on
ships of opportunity while they are underway have amassed throughout the region since the 1970s
(Bakker et al., 2016). Altogether these data, in addition to measurements made by a large
community of agencies, institutions, and stakeholders, constitute an impressive network of
observations that makes Washington one of the most data-rich states to tackle research questions
associated with OA in the coastal zone (Alin et al., 2015).

Building upon these observing efforts, we have characterized the average seasonal cycles of
numerous carbonate system parameters throughout Pacific Northwest marine surface waters.
Comprised of publicly-available, high-quality data, the resulting data products are meant to
provide large-scale environmental context for OA research (Andersson and MacKenzie, 2012;
Hofmann et al., 2011; McElhany and Busch, 2013; Takeshita et al., 2015; Wahl et al., 2016),
adaptation and management strategy development (Boehm et al., 2015; Ekstrom et al., 2015), and
water quality assessment (Bednaršek et al., 2017; Weisberg et al., 2016). Here we describe our
methodology and the resulting carbonate chemistry seasonal cycles, reproduced in the netCDF
format as supplementary material to this article (contact the author for similar information in the
MATLAB format).

**2.  Data sources**
***2.1. Surface Ocean $CO_2$ Atlas version 4***
The Surface Ocean $CO_2$ Atlas (SOCAT; http://www.socat.info/) is the result of a data synthesis
activity carried out by >100 members of the carbon research community who compile and quality
control sea surface (upper ~5 m) measurements of $CO_2$ fugacity ($f$CO$_2$) made across the globe
(Bakker et al., 2016; Pfeil et al., 2013; Sabine et al., 2013), where $f$CO$_2$ is similar to $p$CO$_2$ but takes
into account the non-ideal nature of the gas (Dickson et al., 2007). The SOCAT database is updated
annually to include the most recent observations as well as historical data files that have been
discovered and validated. For this analysis, SOCAT version 4 (SOCAT-v4) $f$CO$_2$ data collected
within 45.5° N to 49° N and 127° W to 122° W (**Fig. 1a**) were downloaded from the coastal
SOCAT database (**Table 1**) along with the accompanying sea surface salinity (SSS) and sea



surface temperature (SST) observations. World Ocean Circulation Experiment quality control
flags of 2 ("good") and SOCAT metadata flags of A through D were applied such that the $f$CO$_2$
observations included have an accuracy of 5 µatm or better (Bakker et al., 2016). Mooring
observations found within the study region were removed from the SOCAT-v4 dataset and are
treated separately. The resultant dataset includes ~57,000 quality-controlled $f$CO$_2$ observations
collected between 1976 and 2015 (**Table 2**). Because these observations span four decades, a
seawater anthropogenic $p$CO$_2$ trend of 1.5 µatm yr$^{-1}$ was used to normalize the $f$CO$_2$ observations
to the reference year 2010, following *Takahashi et al.*, [2009, 2014]. Although a trend in $p$CO$_2$
rather than $f$CO$_2$ was used, the difference in trend between the two parameters is negligible over a
40-year period (~0.006 µatm yr$^{-1}$), resulting in a 0.2 µatm absolute difference between parameters.

***2.2. Discrete carbon data***
Discrete samples collected from the top 10 m of water during recurring oceanographic cruises
conducted between May 2007 and May 2015 near Washington State were compiled for this
analysis (**Fig. 1a**). These cruises include: NOAA Ocean Acidification Program (OAP) West Coast
Ocean Acidification (WCOA) cruises; Pacific Coast Ocean Observing System (PacOOS) cruises;
and UW cruises associated with the Puget Sound Regional Synthesis Model (PRISM) program,
Washington Ocean Acidification Center (WOAC), and Northwest Association of Networked
Ocean Observing Systems (NANOOS) as well as its Chá bǎ mooring. Additionally, ECY collected
discrete water samples via seaplane from ~5 m depth during 2014 and 2015 (Keyzers, 2014, 2016).
Each data subset used herein can be accessed online using information from **Table 1**.

Discrete samples with quality control flags of 2 or 6 (2 = good data, 6 = replicate samples)
were included, yielding ~900 observations. **Table 2** provides a detailed list of the cruises, number
of dissolved inorganic carbon (DIC), total alkalinity (TA), total pH (pH$_T$), $f$CO$_2$, salinity (S), and
temperature (T) observations relied on from each cruise. Information about the silicate and
phosphate observations that were also used in this analysis can be found in **Appendix A**. All of
the DIC and TA bottle samples were analyzed at the Pacific Marine Environmental Laboratory via
coulometric (Dickson et al., 2007; Johnson et al., 1998) and potentiometric titration (Dickson et
al., 2007; Millero et al., 1993), respectively. For all cruises, the accuracy of DIC and TA relative
to Certified Reference Materials (Dickson et al., 2007) is ±0.1 % of the measurement value and



the precision is < ±0.1 % for TA and ~1 µmol kg$^{-1}$ for DIC. This is equal to an accuracy of
approximately 2 µmol kg$^{-1}$ for both parameters in Washington surface waters. For simplicity, we
use 2 µmol kg$^{-1}$ as the total uncertainty for DIC and TA samples and discuss the role of
measurement error further in Sect. 4.2. pH$_T$ measurements were also made during the 2011 and
2013 WCOA cruises (**Table 2**). These samples were analyzed by spectrophotometry (Byrne et al.,
2010; Liu et al., 2011) and have measurement accuracies of 0.004.



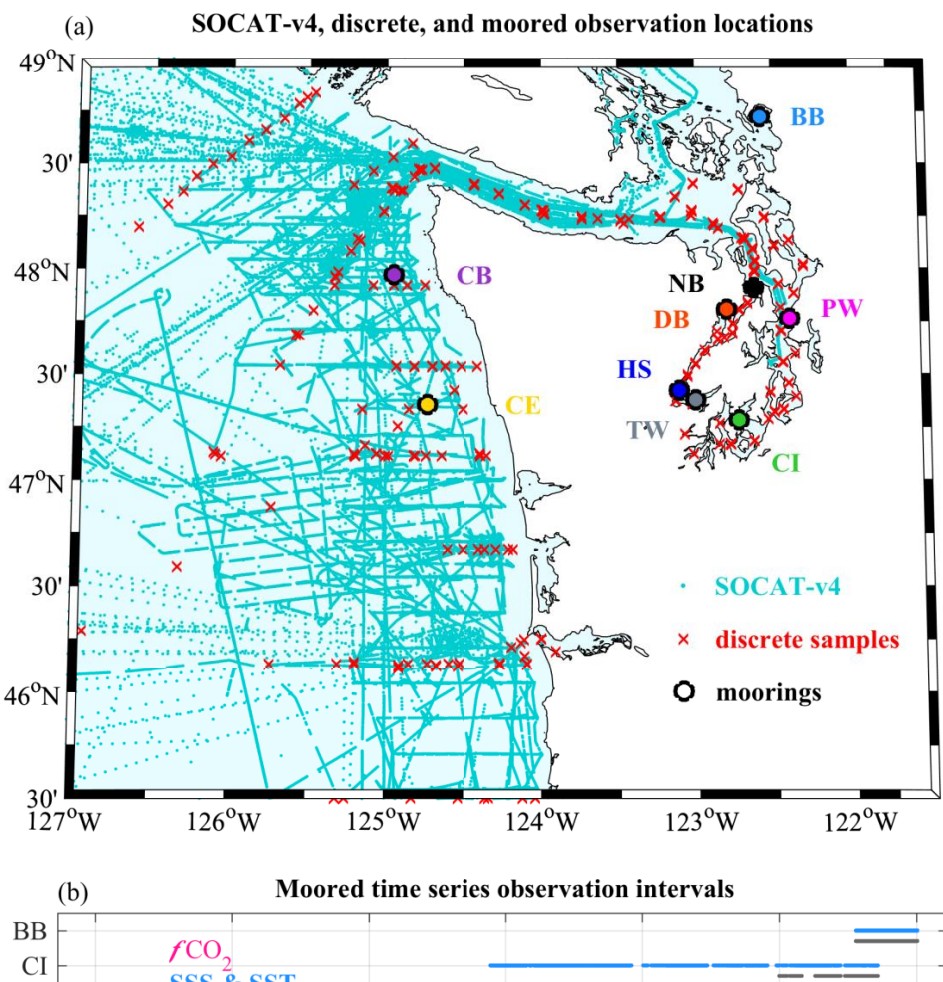

**Figure 1: (a)** Locations of coastal SOCAT-v4 $f$CO$_2$ observations, discrete sample stations, and moorings. The moorings are labeled with two-letter acronyms as follows: Bellingham Bay (BB), Carr Inlet (CI), Dabob Bay (DB), Chá bă (CB), Point Wells (PW), Cape Elizabeth (CE), North Buoy (NB), Hoodsport (HS), and Twanoh (TW). **(b)** Intervals of available sensor data from each of the nine moorings. All moorings include salinity (SSS) and temperature (SST) observations (blue). Some moorings also include pH (gray) and/or $f$CO$_2$ (pink) observations.



**Table 1**. Data attribution table listing the SOCAT and discrete datasets (with observing periods)
incorporated into the analysis. Data curating organization URLs and data DOIs are provided.

| DISCRETE DATA | Obs. Period | Data Curating Organization URL | Data DOI |
|---|---|---|---|
| SOCAT-v4[1] | 1970-2015 | https://doi.pangaea.de/10.1594/PANGAEA.866876 | https://doi.pangaea.de/10.1594/PANGAEA.866856 |
| NOAA WCOA | 05/2007 | https://www.nodc.noaa.gov/ocads/data/0083685.xml | 10.3334/CDIAC/otg.CLIVAR_NACP_West_Coast_Cruise_2007 |
| UW PRISM* | 02/2008 | http://nvs.nanoos.org/CruiseSalish | https://doi.org/10.5281/zenodo.1184657 |
| UW PRISM/EPA* | 08/2008 | http://nvs.nanoos.org/CruiseSalish | https://doi.org/10.5281/zenodo.1184657 |
| PacOOS* | 08/2009 | | https://doi.org/10.5281/zenodo.1184657 |
| UW PRISM* | 09/2009 | http://nvs.nanoos.org/CruiseSalish | https://doi.org/10.5281/zenodo.1184657 |
| PacOOS* | 05/2010 | | https://doi.org/10.5281/zenodo.1184657 |
| PacOOS* | 08/2010 | | https://doi.org/10.5281/zenodo.1184657 |
| UW PRISM* | 11/2010 | http://nvs.nanoos.org/CruiseSalish | https://doi.org/10.5281/zenodo.1184657 |
| UW/Chá bă* | 05/2011 | http://nvs.nanoos.org/CruiseSalish | https://doi.org/10.5281/zenodo.1184657 |
| NOAA WCOA | 08/2011 | https://www.nodc.noaa.gov/ocads/data/0157458.xml | 10.3334/CDIAC/OTG.COAST_WCOA2011 |
| UW PRISM* | 10/2011 | http://nvs.nanoos.org/CruiseSalish | https://doi.org/10.5281/zenodo.1184657 |
| UW/NANOOS* | 05/2012 | http://nvs.nanoos.org/CruiseSalish | https://doi.org/10.5281/zenodo.1184657 |
| NOAA WCOA | 09/2012 | https://www.nodc.noaa.gov/ocads/data/0157445.xml | 10.3334/CDIAC/OTG.COAST_WCOA2012 |
| UW/Chá bă* | 01/2013 | http://nvs.nanoos.org/CruiseSalish | https://doi.org/10.5281/zenodo.1184657 |
| UW/NANOOS* | 04/2013 | http://nvs.nanoos.org/CruiseSalish | https://doi.org/10.5281/zenodo.1184657 |
| NOAA WCOA | 08/2013 | https://www.nodc.noaa.gov/oceanacidification/data/0132082.xml | 10.7289/V5C53HXP |
| UW/NANOOS* | 09/2013 | http://nvs.nanoos.org/CruiseSalish | https://doi.org/10.5281/zenodo.1184657 |
| UW/NANOOS* | 06/2014 | http://nvs.nanoos.org/CruiseSalish | https://doi.org/10.5281/zenodo.1184657 |
| ECY | 06/2014 | https://fortress.wa.gov/ecy/publications/SummaryPages/1603032.html | https://doi.org/10.5281/zenodo.1184657 |
| WOAC* | 07/2014 | http://nvs.nanoos.org/CruiseSalish | https://doi.org/10.5281/zenodo.1184657 |
| ECY | 07/2014 | https://fortress.wa.gov/ecy/publications/SummaryPages/1603032.html | https://doi.org/10.5281/zenodo.1184657 |
| ECY | 08/2014 | https://fortress.wa.gov/ecy/publications/SummaryPages/1603032.html | https://doi.org/10.5281/zenodo.1184657 |
| WOAC* | 09/2014 | http://nvs.nanoos.org/CruiseSalish | https://doi.org/10.5281/zenodo.1184657 |
| ECY | 09/2014 | https://fortress.wa.gov/ecy/publications/SummaryPages/1603032.html | https://doi.org/10.5281/zenodo.1184657 |
| UW/Chá bă* | 10/2014 | http://nvs.nanoos.org/CruiseSalish | https://doi.org/10.5281/zenodo.1184657 |
| ECY | 10/2014 | https://fortress.wa.gov/ecy/publications/SummaryPages/1603032.html | https://doi.org/10.5281/zenodo.1184657 |
| ECY | 11/2014 | https://fortress.wa.gov/ecy/publications/SummaryPages/1603032.html | https://doi.org/10.5281/zenodo.1184657 |
| ECY | 01/2015 | https://fortress.wa.gov/ecy/publications/SummaryPages/1603032.html | https://doi.org/10.5281/zenodo.1184657 |
| ECY | 02/2015 | https://fortress.wa.gov/ecy/publications/SummaryPages/1603032.html | https://doi.org/10.5281/zenodo.1184657 |
| ECY | 03/2015 | https://fortress.wa.gov/ecy/publications/SummaryPages/1603032.html | https://doi.org/10.5281/zenodo.1184657 |
| ECY | 04/2015 | https://fortress.wa.gov/ecy/publications/SummaryPages/1603032.html | https://doi.org/10.5281/zenodo.1184657 |
| ECY | 05/2015 | https://fortress.wa.gov/ecy/publications/SummaryPages/1603032.html | https://doi.org/10.5281/zenodo.1184657 |

[1]The coastal subset of SOCAT-v4 data can be easily accessed here: https://www.socat.info/index.php/version-4/
*Full dataset in preparation for submission to a long-term data repository.



**Table 2**. SOCAT and discrete datasets incorporated into the analysis. The observing period and the number of $f$CO$_2$, DIC, TA, pH, S, and T values that were observed (Obs.), calculated (Calc.), and estimated (Est.) for each cruise are provided.

| DISCRETE DATA | Obs. Period | $f$CO$_2$ Obs. | Calc. | DIC Obs. | Calc. | TA Obs. | Est. | pH$_T$ Obs. | Calc. | S Obs. | T Obs. |
|---|---|---|---|---|---|---|---|---|---|---|---|
| SOCAT-v4 | 1976-2015 | 57,425 | 0 | 0 | 52,521 | 0 | 52,524 | 0 | 52,521 | 52,528 | 57,422 |
| NOAA WCOA | 05/2007 | 0 | 11 | 11 | 0 | 11 | 0 | 0 | 11 | 11 | 11 |
| UW PRISM | 02/2008 | 0 | 80 | 80 | 0 | 75 | 5 | 0 | 80 | 80 | 80 |
| UW PRISM/EPA | 08/2008 | 0 | 122 | 122 | 0 | 120 | 2 | 0 | 122 | 122 | 122 |
| PacOOS | 08/2009 | 0 | 8 | 8 | 0 | 8 | 0 | 0 | 8 | 8 | 8 |
| UW PRISM | 09/2009 | 0 | 38 | 38 | 0 | 38 | 0 | 0 | 38 | 38 | 38 |
| PacOOS | 05/2010 | 0 | 13 | 13 | 0 | 13 | 0 | 0 | 13 | 13 | 13 |
| PacOOS | 08/2010 | 0 | 22 | 22 | 0 | 22 | 0 | 0 | 22 | 22 | 22 |
| UW PRISM | 11/2010 | 0 | 40 | 40 | 0 | 37 | 3 | 0 | 40 | 40 | 40 |
| UW/Chá bă | 05/2011 | 0 | 1 | 1 | 0 | 1 | 0 | 0 | 1 | 1 | 1 |
| NOAA WCOA | 08/2011 | 0 | 37 | 37 | 0 | 32 | 5 | 32 | 5 | 37 | 37 |
| UW PRISM | 10/2011 | 0 | 60 | 60 | 0 | 60 | 0 | 0 | 60 | 60 | 60 |
| UW/NANOOS | 05/2012 | 0 | 4 | 4 | 0 | 4 | 0 | 0 | 4 | 4 | 4 |
| NOAA WCOA | 09/2012 | 0 | 43 | 43 | 0 | 39 | 3 | 0 | 43 | 43 | 43 |
| UW/Chá bă | 01/2013 | 0 | 2 | 2 | 0 | 2 | 0 | 0 | 2 | 2 | 2 |
| UW/NANOOS | 04/2013 | 0 | 19 | 19 | 0 | 18 | 1 | 0 | 19 | 19 | 19 |
| NOAA WCOA | 08/2013 | 0 | 59 | 59 | 0 | 47 | 9 | 45 | 14 | 59 | 59 |
| UW/NANOOS | 09/2013 | 0 | 19 | 19 | 0 | 19 | 0 | 0 | 19 | 19 | 19 |
| UW/NANOOS | 06/2014 | 0 | 10 | 10 | 0 | 10 | 0 | 0 | 10 | 10 | 10 |
| ECY | 06/2014 | 0 | 17 | 17 | 0 | 17 | 0 | 0 | 17 | 17 | 17 |
| WOAC | 07/2014 | 0 | 68 | 68 | 0 | 68 | 0 | 0 | 68 | 68 | 68 |
| ECY | 07/2014 | 0 | 12 | 12 | 0 | 12 | 0 | 0 | 12 | 12 | 12 |
| ECY | 08/2014 | 0 | 18 | 18 | 0 | 18 | 0 | 0 | 18 | 18 | 18 |
| WOAC | 09/2014 | 0 | 64 | 64 | 0 | 64 | 0 | 0 | 64 | 64 | 64 |
| ECY | 09/2014 | 0 | 14 | 14 | 0 | 14 | 0 | 0 | 14 | 14 | 14 |
| UW/Chá bă | 10/2014 | 0 | 19 | 19 | 0 | 19 | 0 | 0 | 19 | 19 | 19 |
| ECY | 10/2014 | 0 | 9 | 9 | 0 | 9 | 0 | 0 | 9 | 9 | 9 |
| ECY | 11/2014 | 0 | 9 | 9 | 0 | 9 | 0 | 0 | 9 | 9 | 9 |
| ECY | 01/2015 | 0 | 13 | 13 | 0 | 13 | 0 | 0 | 13 | 13 | 13 |
| ECY | 02/2015 | 0 | 15 | 15 | 0 | 15 | 0 | 0 | 15 | 15 | 15 |
| ECY | 03/2015 | 0 | 20 | 20 | 0 | 20 | 0 | 0 | 20 | 20 | 20 |
| ECY | 04/2015 | 0 | 15 | 15 | 0 | 15 | 0 | 0 | 15 | 15 | 15 |
| ECY | 05/2015 | 0 | 6 | 6 | 0 | 6 | 0 | 0 | 6 | 6 | 6 |
| **TOTAL** | | **57,425** | **887** | **887** | **52,521** | **855** | **52,552** | **77** | **53,331** | **53,415** | **58,309** |

### *2.3. Mooring carbon data*

In addition to discrete data, time-series observations from nine surface moorings in the region are included in the analysis (**Fig. 1a**). These moorings are maintained by NOAA, UW, WOAC, and NANOOS. Moored time-series data can be accessed online using information from **Table 3**. All of the moorings carry sensors that measure SST and SSS, and most include sensors for surface



ocean $f\mathrm{CO_2}$ or $\mathrm{pH_T}$, while the Chá bă mooring measures both of these carbon parameters (**Fig. 1b**).
Presently, quality-controlled observations from the moorings range from January 2005 through
January 2017, with the longest carbon record from the Cape Elizabeth mooring that started in 2006.
Assuming the same anthropogenic $f\mathrm{CO_2}$ trend of 1.5 µatm yr$^{-1}$ that was used to normalize the
SOCAT-v4 data (*Takahashi et al.*, 2009, 2014), the Cape Elizabeth time series may include a ~12
µatm anthropogenic increase in the mean $f\mathrm{CO_2}$ over the 8-year record. This is much smaller than
the potential ~60 µatm increase over the ~40 year SOCAT-v4 record. Additionally, since all other
moorings have much shorter carbon time series (some with pH rather than $f\mathrm{CO_2}$ records), we do
not remove the anthropogenic $f\mathrm{CO_2}$ trend from the mooring data.

The Chá bă, Cape Elizabeth, Dabob Bay, and Twanoh surface moorings each carry a Moored
Autonomous $p\mathrm{CO_2}$ (MAPCO2) system that makes 3-hour measurements of surface air and sea
$p\mathrm{CO_2}$ with an accuracy of $\leq 2$ µatm (Sutton et al., 2014). The Chá bă mooring is also outfitted with
a Sunburst SAMI pH sensor with reported field accuracy of 0.018, and a Sea-Bird Electronics
(SBE) 16plus V2 Sea-CAT conductivity-temperature-depth (CTD) sensor with SST accuracy <
0.01 and SSS accuracy < 0.05, as described in Sutton et al. (2016). The Cape Elizabeth mooring
carries an SBE 37 MicroCAT CTD sensor with SST accuracy < 0.01 and SSS accuracy < 0.05.
These sensors are programed to sample every 3 hours in tandem with the MAPCO2 system. The
other seven moorings, Oceanic Remote Chemical/Optical Analyzer (ORCA) buoys (Dunne et al.,
2002), are equipped with profiling sensors that include SBE19 or 19Plus CTD sensors (SST
accuracy of 0.005 and SSS accuracy of 0.005), SBE 43 dissolved oxygen sensors, and WETLabs
WETStar or FLNTUS chlorophyll fluorometers. Temperature and salinity observations collected
from the top 3 m of the water column were binned and averaged to serve as SST and SSS for this
analysis. The Carr Inlet and Bellingham Bay surface buoys also carry Satlantic SeaFET pH sensors
with a reported accuracy of ± 0.05 pH. All raw SeaFET measurements were converted to pH using
custom code adapted from Martz et al. (2015) and equations from Bresnahan et al. (2014). ORCA
mooring data were quality controlled according to protocols outlined in the Quality Assurance
Project Plan (http://nwem.ocean.washington.edu/ORCA_QAPP.pdf; Newton and Devol, 2012).
All sensors on the surface buoys sample the water column at ~1 m depth, excluding the profiling
CTDs. Details about the number of observations collected at each mooring location are provided
in **Table 4**. It is important to note that many of the moored time series include significant data gaps





and/or times when $f$CO$_2$ and pH$_T$ measurements are not available (**Fig. 1b**). The observing periods
listed in **Table 3** represent the time range over which quality-controlled, *in situ* observations exist
at the mooring site at the time of this manuscript.

**Table 3**. Data attribution table listing the mooring datasets (with observing periods) incorporated
into the analysis. Data curating organization URLs and data DOIs are provided.

| MOORING DATA | Obs. Period | Data Curating Organization URL | Data DOI |
|---|---|---|---|
| Chá bǎ[1] | 07/10-10/15 | https://www.nodc.noaa.gov/ocads/data/0100072.xml | https://doi.org/10.3334/CDIAC/OTG.TSM_LAPUSH_125W_48N |
| Cape Elizabeth | 06/06-09/13 | https://www.nodc.noaa.gov/ocads/data/0115322.xml | https://doi.org/10.3334/CDIAC/OTG.TSM_WA_125W_47N |
| Dabob Bay* | 06/10-11/14 | https://www.nodc.noaa.gov/ocads/data/0116715.xml | https://doi.org/10.3334/CDIAC/OTG.TSM_DABOB_122W_478N |
| Twanoh* | 01/05-06/16 | https://www.nodc.noaa.gov/ocads/data/0157600.xml | https://doi.org/10.3334/CDIAC/OTG.TSM_TWANOH_123W_47N |
| Carr Inlet | 10/10-06/16 | http://nwem.ocean.washington.edu/prod_Data_Req.shtml | https://doi.org/10.5281/zenodo.1184657 |
| Bellingham Bay | 02/16-01/17 | http://nwem.ocean.washington.edu/prod_Data_Req.shtml | https://doi.org/10.5281/zenodo.1184657 |
| Hoodsport | 10/05-05/16 | http://nwem.ocean.washington.edu/prod_Data_Req.shtml | https://doi.org/10.5281/zenodo.1184657 |
| Point Wells | 04/10-06/16 | http://nwem.ocean.washington.edu/prod_Data_Req.shtml | https://doi.org/10.5281/zenodo.1184657 |
| North Buoy | 11/05-05/16 | http://nwem.ocean.washington.edu/prod_Data_Req.shtml | https://doi.org/10.5281/zenodo.1184657 |

[1]The Chá bǎ mooring is referred to as the La Push mooring in the NOAA data repository.
*Temperature and salinity data for the Dabob Bay and Twanoh moorings can be found here:
https://doi.org/10.5281/zenodo.1184657

**Table 4**. Mooring datasets incorporated into the analysis. The observing period and number of
$f$CO$_2$, DIC, TA, pH, S, and T values that were observed (Obs.), calculated (Calc.), and estimated
(Est.) are provided.

| MOORING DATA | Obs. Period | $f$CO$_2$ Obs. | $f$CO$_2$ Calc. | DIC Obs. | DIC Calc. | TA Obs. | TA Est. | pH$_T$ Obs. | pH$_T$ Calc. | S Obs. | T Obs. |
|---|---|---|---|---|---|---|---|---|---|---|---|
| Chá bǎ | 07/10-11/15 | 9,681 | 0 | 0 | 9,330 | 0 | 9,341 | 7,438 | 2,291 | 9,341 | 9,717 |
| Cape Elizabeth | 06/06-03/15 | 17,617 | 0 | 0 | 17,617 | 0 | 17,667 | 0 | 17,617 | 17,667 | 18,009 |
| Dabob Bay | 06/10-11/14 | 2,076 | 0 | 0 | 2,076 | 0 | 4,662 | 0 | 2,076 | 4,662 | 4,662 |
| Twanoh | 01/05-06/16 | 9,221 | 0 | 0 | 9,221 | 0 | 28,466 | 0 | 9,221 | 28,466 | 28,466 |
| Carr Inlet | 10/10-06/16 | 0 | 1,848 | 0 | 1,848 | 0 | 12,369 | 1,848 | 0 | 12,369 | 12,369 |
| Bellingham Bay | 02/16-01/17 | 0 | 0 | 0 | 0 | 0 | 0 | 45,583 | 0 | 44,895 | 44,895 |
| Hoodsport | 10/05-05/16 | 0 | 0 | 0 | 0 | 0 | 17,147 | 0 | 0 | 17,186 | 17,186 |
| Point Wells | 04/10-06/16 | 0 | 0 | 0 | 0 | 0 | 4,315 | 0 | 0 | 4,315 | 4,315 |
| North Buoy | 11/05-05/16 | 0 | 0 | 0 | 0 | 0 | 10,199 | 0 | 0 | 10,199 | 10,199 |
| **TOTAL** | | **38,595** | **1,848** | **0** | **40,092** | **0** | **104,166** | **54,869** | **31,205** | **149,100** | **149,818** |


## 3. Methods
### 3.1. Leveraging the carbon datasets



Seawater carbonate chemistry can be fully characterized with information about salinity,
temperature, pressure, weak acids that contribute to TA (e.g., phosphoric and silicic acid), and two
of the four commonly-measured carbonate system parameters: DIC, TA, $pCO_2$ (or $fCO_2$), and pH
(Dickson and Riley, 1978; Millero, 2007). In order to leverage the numerous SOCAT-v4 $fCO_2$
observations that lack a second carbonate system parameter, a previously characterized
relationship between near-surface ($\leq 25$ m) salinity and TA (TA = $47.7 \times$ Salinity + 647) for marine
surface waters near Washington (Fassbender et al., 2017a) was used to estimate TA from the
SOCAT-v4 SSS observations. This relationship yields TA estimates that have an uncertainty $\leq$
$\pm34$ µmol kg$^{-1}$ at the 95% confidence level ($2\sigma$) for waters with salinity $\geq 20$ (Fassbender et al.,
2017). Uncertainties may be larger in the salinity range of 20 to 27 ($2\sigma = \pm94$ µmol kg$^{-1}$); however,
only ~2% of the SOCAT-v4 SSS observations are below salinity 27. TA was not estimated when
SOCAT-v4 SSS values were below 20 (n=4). The estimated TA and 2010-normalized SOCAT-v4
$fCO_2$ data were then used to calculate other carbonate system parameters of interest ("calculated"
values in **Table 2**). All carbonate system calculations herein were made using the program
CO2SYS Version 1.1 (van Heuven et al., 2011; Lewis and Wallace, 1998) with the equilibrium
constants of Lueker et al. (2000) and Dickson (1990) while applying the boron-to-chlorinity ratio
of Uppstrom, (1974), following the recommendations of Orr et al. (2015). SOCAT-v4 does not
include nutrient observations, so all of the associated CO2SYS calculations were made with
phosphate and silicate concentrations set to zero. The omission of nutrients when using the TA-
$fCO_2$ pair to calculate other carbonate system parameters results in small errors. These errors were
previously assessed for the TA-$pCO_2$ pair within Washington waters (Fassbender et al., 2017a),
and will be discussed further in Sect. 3.2.

TA and DIC were measured during discrete sampling efforts in addition to temperature,
salinity, and pressure, and these data were used to calculate other carbonate system parameters of
interest. Some of the discrete sample datasets lack nutrient observations, so phosphate and silicate
concentration were set to zero for the associated CO2SYS calculations. Nutrient omission when
using the TA-DIC pair has a larger influence on calculated carbonate system parameters than the
TA-$pCO_2$ pair (Fassbender et al., 2017), which will be addressed further in Sect. 3.2. In some
cases, only DIC was measured. For these instances, TA was estimated from the previously
mentioned regional TA-salinity relationship when salinity was $\geq 20$ (true of all discrete samples),





making it possible to fully characterize the carbonate system as long as DIC, SST, and SSS data
were available. Finally, on cruises in which $pH_T$ was measured directly, missing $pH_T$ observations
were calculated from TA and DIC. Details about the specific number of measured, estimated, and
calculated values for each parameter can be found in **Table 2**.

Sustained, autonomous $f$CO$_2$ and pH measurements made on some of the surface moorings
located throughout the region (**Fig. 1a**) were also used to evaluate carbonate chemistry. The TA-
salinity relationship was applied to the moored SSS observations to estimate TA for pairing with
the *in situ* $f$CO$_2$ or pH data (**Fig. 1b**; **Table 4**), allowing calculation of other carbonate system
parameters of interest. For the Chá bă mooring, where $f$CO$_2$ and pH measurements have been made
simultaneously since 2010, $f$CO$_2$ and TA estimates were used to calculate additional carbonate
system parameters due to challenges associated with constraining the carbonate system with the
$f$CO$_2$-pH pair (Dickson and Riley, 1978; Fassbender et al., 2015; Gray et al., 2011). At moorings
limited to SSS and SST observations, TA was the only carbonate system parameter determined.
Importantly, salinity at the Twanoh and Hoodsport moorings occasionally dropped below 20
throughout the observational record, and salinity at the Bellingham Bay mooring was often < 20.
These samples were not used to estimate TA. Because the occurrence of salinity < 20 was
infrequent at Twanoh and Hoodsport (1.7% and 0.2% of samples, respectively), the exclusion of
these observations from seasonal cycle evaluations likely has a negligible impact. At Bellingham
Bay, regular occurrences of salinity < 20 (14% of samples) throughout all seasons renders the TA-
salinity relationship unusable for a substantial fraction of the time series. Therefore, we do not
calculate TA, or any other carbonate system parameters, and evaluate only *in situ* $pH_T$ observations
from the Bellingham Bay mooring.

For this study, the carbonate system parameters of interest (either measured, estimated, or
calculated) include: TA, DIC, $pH_T$, $f$CO$_2$, aragonite saturation state ($\Omega_{Ar}$), and the Revelle Factor
(RF). $\Omega_{Ar}$ describes the thermodynamic stability of aragonite in solution, where aragonite is a
polymorph of the mineral calcium carbonate (CaCO$_3$). This stability is dependent on temperature,
pressure, salinity, and the concentrations of calcium and carbonate ions (Mucci, 1983). By
definition, when $\Omega_{Ar} \geq 1$, the mineral is thermodynamically stable. When $\Omega_{Ar} < 1$, the mineral is
unstable and will begin to dissolve. This is relevant to the study of OA because many marine



calcifying organisms make their shells out of aragonite and are susceptible to dissolution when
exposed to waters with low (often $\Omega_{Ar} < 1$, but occasionally higher) aragonite saturation states
(Barton et al., 2012; Bednaršek et al., 2014; Doney et al., 2009; Feely et al., 2008). The Revelle
Factor is analogous to the buffer capacity of the ocean (Broecker et al., 1979; Revelle and Suess,
1957; Sundquist et al., 1979; Takahashi et al., 1980) and provides information about how the
carbonate system responds to change. Formally, RF is equal to the fractional change in $p$CO$_2$ (or
$f$CO$_2$) divided by the fractional change in DIC resulting from a given perturbation. Lower RF
values equate to more buffered systems and result in a larger DIC change per percentage increase
in $p$CO$_2$ due to more efficient conversion of CO$_2$ into other molecular forms of DIC (e.g.,
bicarbonate and carbonate ion), which enables additional seawater CO$_2$ uptake. Thus, low RF
regions are associated with larger anthropogenic carbon uptake through air-sea exchange relative
to high RF regions (e.g., Fassbender et al., 2017b; Sabine et al., 2004).

### 3.2. Errors associated with nutrient omission and estimating TA from salinity

Silicate and phosphate contribute to TA (Dickson, 1981) and when omitted from CO2SYS
carbonate system calculations can have a small but non-negligible influence on the calculated
parameters. This was evaluated previously by Fassbender et al. (2017a) for many of the cruise data
used herein; however, RF was not included in the analysis and the TA-$p$CO$_2$ pair was used rather
than TA-$f$CO$_2$ pair. Observations from regional cruises (**Table 2**) that include complete
information about TA, DIC, phosphate, silicate, salinity, and temperature were used to calculate
carbonate system parameters of interest using the observed nutrient concentrations and with
nutrient concentrations set to zero. The analysis was conducted with both the TA-DIC pair and the
TA-$f$CO$_2$ pair as input parameters, where $f$CO$_2$ values were initially calculated from TA and DIC
using the observed nutrient concentrations. Differences between the computations made with and
without nutrients are given in **Table 5**, displaying the larger influence of nutrient omission on
calculations made from the TA-DIC pair than the TA-$f$CO$_2$ pair for pH and $\Omega_{Ar}$, though not for RF
due to its strong dependence to the DIC-TA ratio.

In addition to missing nutrient data, there is uncertainty in the Washington TA-salinity
relationship (Fassbender et al., 2017a), which characterizes a static mean condition from which
deviations can occur in both space and time. Biases in the estimated TA value caused by departures



from the mean condition may therefore create biases in the carbonate system parameters derived
from it. This is of particular relevance in regard to leveraging the SOCAT-v4 and mooring datasets
due to the explicit use of estimated TA to constrain the carbonate system. To evaluate the
magnitude of these potential biases, the same ship data discussed in the previous paragraph were
used (with nutrient concentrations set to zero) to compute carbonate system parameters of interest
from the TA-$f$CO$_2$ pair after adding and subtracting the $\pm 2\sigma$ TA-salinity regression uncertainty (34
µmol kg$^{-1}$) to the TA values. Absolute differences between the resulting values and original
estimates from the TA-$f$CO$_2$ pair (without altering the TA) were then averaged (**Table 5**). The
errors associated with using the Washington TA-salinity relationship are similar in magnitude to
those caused by nutrient omission when using the DIC-TA pair and much larger than those
resulting from nutrient omission when using the TA- $f$CO$_2$ pair, excluding for RF. These results
suggest that $\Omega_{Ar}$ is more sensitive to the TA-salinity regression uncertainty than nutrient omission
and RF is more sensitive to nutrient omission in the TA-$f$CO$_2$ pair than uncertainty in TA-salinity
regression within Washington waters. Notably, most of the errors are small relative to sensor
accuracies and/or natural variability (including for $f$CO$_2$ and DIC), which will be addressed in Sect.

4.2.


**Table 5.** Errors associated with calculating carbonate system parameters in the absence of nutrient
data and when using salinity-based estimates of TA for marine surface waters near Washington.
Results for the TA-DIC and TA-$f$CO$_2$ pairs of input parameters are shown.

| OUTPUT → <br> INPUT ↓ | pH | $\Omega_{Ar}$ | RF | $f$CO$_2$ <br> (µatm) | DIC <br> (µmol kg$^{-1}$) |
|---|---|---|---|---|---|
| **TA & DIC (w/ - w/o Nutrients)** | -0.007 | -0.02 | -0.009 | 11 | - |
| **TA & $f$CO$_2$ (w/ - w/o Nutrients)** | -0.0004 | -0.002 | -0.07 | - | -2 |
| **TA & $f$CO$_2$ (±2σ on Estimated TA)** | ±0.007 | ±0.05 | ±0.02 | - | ±30 |


Finally, SOCAT-v4 and mooring SSS data used herein were not rigorously quality controlled
beyond the identification of large outliers. Because TA is estimated directly from these salinity
observations, we performed a simple evaluation of the TA-salinity regression sensitivity. Modern,
generic conductivity sensors can achieve salinity measurements with an accuracy of ~0.02. To be
conservative, we propagate a salinity error ten times this value (±0.2) through the regression
calculations. This yields a mean error of ±10 µmol kg$^{-1}$ in TA, which is well within the TA
regression 95% confidence interval of ±34 µmol kg$^{-1}$. In order to exceed the regression uncertainty,



a salinity error of ~0.7, which is 35 times the presumed accuracy of 0.02, would be required. As a
result, salinity measurement errors are likely secondary to uncertainties associated with the
regression relationship.

*3.3. New data products of monthly averaged carbonate chemistry*

Discrete sample and SOCAT-v4 datasets (including estimated and calculated values) were

combined and gridded to a 0.1°×0.1° scale. From here on, this merged product will be referred to
as the discrete data product. **Figure 2a** shows the total number of times that two carbon parameters
were available to fully characterize the carbonate system within each 0.1°×0.1° grid cell. Monthly
means, unique to each year ($mm_{yr}$), were calculated for each parameter then averaged over the
number of years (n) with data to get the overall mean value ($\overline{mm}$) for each month of the seasonal
cycle.

$$\overline{mm} = \frac{\sum_{yr=1}^{n} mm_{yr}}{n}$$   **(1)**

The same approach was applied to the mooring data so that monthly observations from each year
were weighted equally, regardless of the number of available samples. Average seasonal cycles of
all parameters for each region and mooring are tabulated in **Appendices B** and **C**.

Monthly variance values, unique to each year ($m\sigma^2_{yr}$), were calculated for each parameter then

averaged over the number of years with data before the square root was taken to determine the
mean standard deviation ($\overline{m\sigma}$) for each month of the seasonal cycle.

$$\overline{m\sigma} = \sqrt{\frac{\sum_{yr=1}^{n} m\sigma^2_{yr}}{n}}$$   **(2)**

**Fig. 3** shows monthly averaged SST data from each annual Cape Elizabeth (CE) mooring
deployment ($mm_{yr}$ values) as well as the multi-year average seasonal cycle ($\overline{mm}$ values). Shading
bounds the mean standard deviation (**Eq. 2**; **Fig. 3a**) and the standard deviation of monthly mean
values (**Fig. 3b**), as expressed in **Eq. 3**:

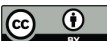



$$\sigma = \sqrt{\frac{\sum_{yr=1}^{n}\left(mm_{yr}-\overline{mm}\right)^2}{n-1}}$$
(3)


The former (**Eq. 2**) provides an estimate of the range of variability about the mean that is observed
in a given month, while the latter (**Eq. 3**) provides an estimate of interannual variability in the
mean. Both methods deliver equally useful information; however, nearly all of the moored time
series are shorter than a decade, which hinders making robust estimates of interannual variability.
Thus, standard deviations presented herein (and tabulated in **Appendix C**) reflect the average
range of variability about the mean during each month (i.e., **Eq. 2**; **Fig. 3a**).

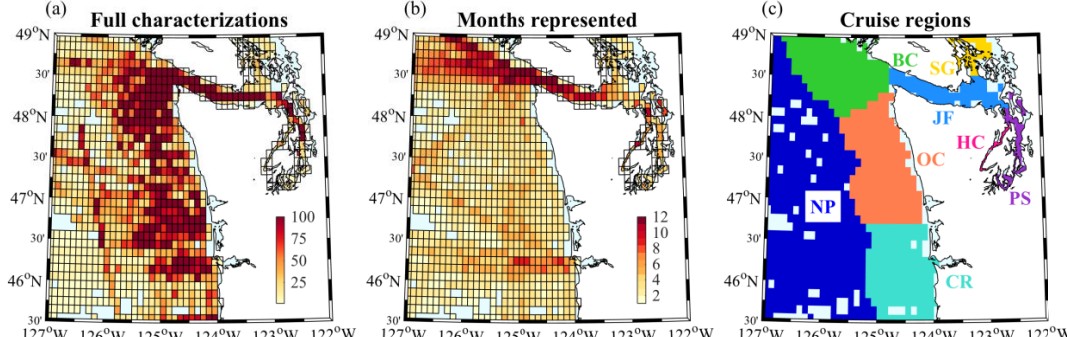


**Figure 2:** Maps showing (**a**) the total number of times the carbonate system was fully characterized
(two parameters available) in each grid cell, (**b**) the number of months represented in the seasonal
cycle for each grid cell, and (**c**) the eight unique regions identified, including: North Pacific (NP),
British Columbia (BC), Outer Coast (OC), Columbia River (CR), Juan de Fuca (JF), Strait of
Georgia (SG), Hood Canal (HC), and Puget Sound (PS).

**Figure 2b** shows the number of months represented in the seasonal cycle estimate for each
grid cell, highlighting that all 12 months are rarely resolved within a single grid cell. It is important
to note, however, that neighboring grid cells with, for example, six months of the year represented
may have observations from different months, making it possible to fully resolve the annual cycle
by combining the seasonal cycles of neighboring grid cells with similar characteristics. Relying on
this approach to accommodate data paucity, salinity and topographic features (e.g., the Juan de
Fuca Canyon and continental shelf) were used to identify eight unique regions (**Fig. 2c**) in which
seasonal cycles from individual grid cells were averaged to create regional estimates. These
regions include: North Pacific (NP), British Columbia coast (BC), Washington Outer Coast (OC),
Columbia River (CR), Strait of Juan de Fuca (JF), Strait of Georgia (SG), Hood Canal (HC), and
Puget Sound (PS). Importantly, some of the major physical features that result in regionally distinct
characteristics, such as the Columbia and Fraser River plumes and the Juan de Fuca Eddy, vary in
extent seasonally. The nonuniformity of available data in space and time means that these features
and their seasonal transitions may not be adequately captured in all grid cells. In addition, the lines
between regional boundaries can be blurred by interannual variability that may be inconsistently
represented in neighboring grid cells. Thus, the dynamic nature of the coastal zone should be
considered when applying the regional seasonal cycles characterized herein, which are presented
in a static-boundary framework. Due to the unequal distribution of discrete observations in space
and time, monthly mean standard deviations for the regional seasonal cycle estimates do not
provide reliable information and are not included in **Appendix B**.

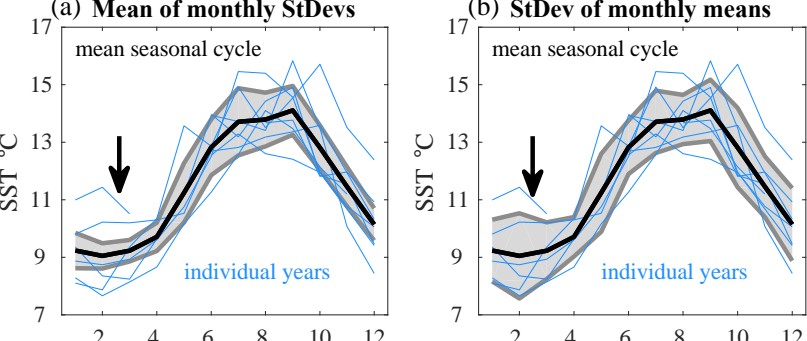


**Figure 3**. Monthly averaged SST values for each year (2006-2015) of Cape Elizabeth mooring
data ($\overline{mm}_{yr}$; blue lines) and the average seasonal cycle of SST across all deployments ($\overline{mm}$; black
line). Blue and black lines are the same in plots **a** and **b**. Gray shading represents (**a**) the mean
monthly standard deviation ($\overline{m\sigma}$) and (**b**) the standard deviation of monthly means ($\sigma$). Black
arrows point to a region of notable discrepancy between computations. Here, the average sub-
monthly SST variability (**a**) is much smaller than the range of interannual SST variability (**b**),
indicating that organisms could experience distinct (i.e., non-overlapping) SST conditions during
two subsequent Februaries.

**4.   Results and discussion**
*4.1.  Modern seasonal cycles*

Seasonal cycles for the eight regions, derived using measurements from the top 10 m of the

water column, are shown in **Fig. 4**. A clear pattern of salinity decrease is observed moving from
the offshore North Pacific region toward land where rivers contribute freshwater to the marine



environment. In particular, the Columbia and Fraser River outflows are evident in the seasonal
cycles for the Columbia River and Strait of Georgia regions, with other small rivers exerting a
strong influence on the semi-enclosed Hood Canal and Puget Sound domains. SST seasonal cycles
are similar in structure across regions, excluding the Strait of Juan de Fuca where vigorous mixing
driven by bottom topography results in much cooler surface temperatures than neighboring areas
throughout the year (Alford and MacCready, 2014; Martin and MacCready, 2011). The mean
seasonal cycles for $f\mathrm{CO_2}$ and pH are approximate mirror images within each region as a result of
the strong negative correlation between these two parameters (Dickson and Riley, 1978). Annual
mean $f\mathrm{CO_2}$ values increase and pH values decline when moving from the offshore North Pacific
toward the Salish Sea, and the seasonal cycle amplitudes increase for both parameters. In all
regions, the lowest $f\mathrm{CO_2}$ values emerge in summer and the highest values in winter, which opposes
expectations associated with temperature-driven solubility changes (Takahashi et al., 1993). This
suggests that physical and biological effects dominate over solubility changes in all regions
(Pelletier et al., 2018), though to varying extents. Due to the logarithmic nature of the pH scale,
declines in the annual mean pH coupled with increases in the seasonal range of pH values means
that organisms in the Salish Sea (Puget Sound, Hood Canal, Strait of Juan de Fuca, and Strait of
Georgia) are exposed to extremely large changes in [H$^+$] on seasonal time scales. For example, the
seasonal range of [H$^+$] in Hood Canal is ~27 times larger than in the North Pacific region, which
indicates that organisms are living in starkly different chemical regimes across a relatively small
spatial domain.

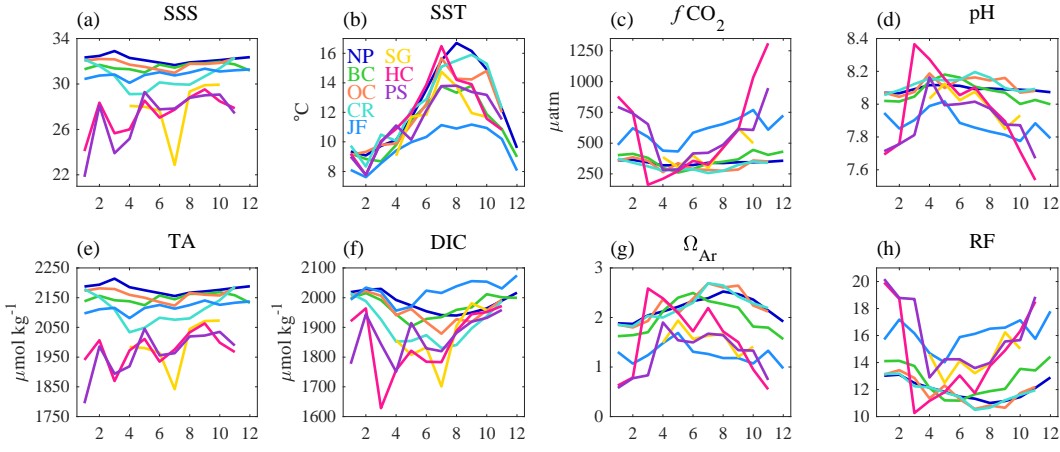



**Figure 4.** Near surface (< 10 m) seasonal cycles of (**a**) SSS, (**b**) SST, (**c**) $f$CO$_2$, (**d**) pH$_T$, (**e**) TA, (**f**) DIC, (**g**) $\Omega_{Ar}$, and (**h**) Revelle Factor (RF) for the eight regions shown in **Fig. 2c**: North Pacific (NP), British Columbia (BC), Outer Coast (OC), Columbia River (CR), Juan de Fuca (JF), Strait of Georgia (SG), Hood Canal (HC), and Puget Sound (PS).

Most regions exhibit moderate seasonal TA variability with the exception of the Strait of Georgia where the Fraser River has a significant impact on the TA. Additionally, the Hood Canal and Puget Sound regions display somewhat erratic seasonal TA variations likely due to sporadic river influences and/or the small number of available samples that are unevenly distributed in space and time (**Appendix B**). DIC seasonal cycles are similar to TA but with larger amplitudes. Generally, DIC and TA are highest during winter and lowest during summer, with DIC:TA being lowest in summer. $\Omega_{Ar}$ is dependent on temperature and the carbonate ion concentration [CO$_3^{2-}$], which is inversely proportional to DIC:TA. $\Omega_{Ar}$ seasonal cycles closely track (DIC:TA)$^{-1}$ changes in all regions, expressing the highest values during summer and the lowest values during winter. If SST is held constant in the CO2SYS calculations of $\Omega_{Ar}$ for each region, the resulting seasonal cycle is altered by <10% for all regions. This indicates that local chemistry changes are controlling the $\Omega_{Ar}$ seasonal cycle throughout the study domain, with temperature contributing minimally. Finally, RF values are elevated during winter and depressed during summer indicating a larger buffer capacity of waters during summer. RF values in the Salish Sea are ~50% higher than those in coastal regions, particularly during winter, indicating that Salish Sea waters are significantly less efficient at absorbing anthropogenic carbon than open ocean waters at the same latitude. In general, seasonal cycle amplitudes are largest within the Salish Sea for all parameters.

Seasonal cycles for the nine moorings, using data collected from ~1 m depth, are shown in **Fig. 5** and generally follow the same pattern as the eight regions. SSS decreases when moving inland from the outer coast toward Puget Sound. SST cycles are similar across moorings, with larger amplitudes found in shallower basins, such as at the Bellingham Bay and Carr Inlet moorings, as well as the moorings in Hood Canal where vertical mixing is weaker than in other parts of Puget Sound (Newton et al., 2003). pH and $f$CO$_2$ display opposing seasonal cycles and TA follows the seasonality of salinity, as expected, exhibiting lower values during spring and summer and higher values during fall and winter. DIC displays more pronounced seasonal cycles with the lowest values in summer and highest values in winter. As seen for the regional discrete sample data, $\Omega_{Ar}$

is highest in summer and lowest in winter following its inverse relationship with DIC:TA. Finally,
RF values are lowest in summer and highest in winter, with the largest annual mean values in the
Salish Sea. In general, seasonal cycle amplitudes for all parameters are higher in the Salish Sea
than on the outer coast, similar to what was found for the regional evaluations.

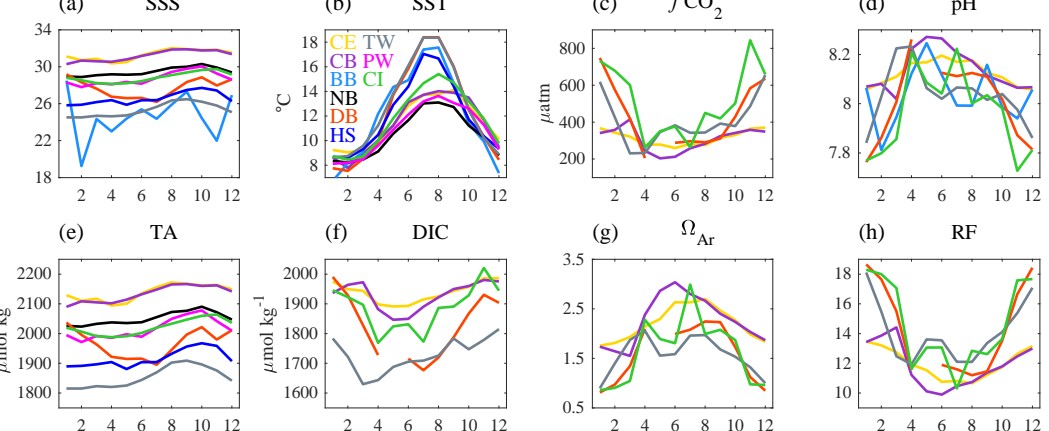

**Figure 5.** Near surface (~1 m) seasonal cycles of (**a**) SSS, (**b**) SST, (**c**) $f$CO$_2$, (**d**) pH$_T$, (**e**) TA, (**f**)
DIC, (**g**) Ω$_{Ar}$, and (**h**) Revelle Factor (RF) for the nine moorings shown in **Fig. 1a**: Cape Elizabeth
(CE), Chá bă (CB), Carr Inlet (CI), Dabob Bay (DB), Twanoh (TW), Hoodsport (HS), Point Wells
(PW), North Buoy (NB), and Bellingham Bay (BB).

A shared feature of the region and mooring seasonal cycle results is the seasonal evolution of
offshore-onshore gradients in numerous parameters. For example, most regions and moorings
exhibit similar winter SST values; however, large gradients in SST are found between sites during
summer. Alternatively, pH, Ω$_{Ar}$, and $f$CO$_2$ are most similar across sites during summer, with the
largest gradients found during winter. These findings suggest that seasonal resolution is needed to
accurately describe the spatial pattern of chemical gradients in the Pacific Northwest, particularly
since different parameters exhibit maximum gradients at dissimilar times of the year. Consistency
between the large-scale seasonal cycle patterns derived from the nine moorings and eight unique
regions is somewhat surprising due to differences in the sampling depth, with the moorings
sampling the top ~1m of the water column and regional seasonal cycles derived from samples
within the top 10 m. Since deeper waters are generally cooler and have higher salinity, $f$CO$_2$, TA,
and DIC values, this implies that seasonal forcing overwhelms any influence of stratification,
yielding the broad agreement between region and mooring seasonal cycles.




To more directly compare results from the regions and moorings and evaluate large-scale
patterns of variability, we focus on seasonal cycles from the Washington Coast, Puget Sound, and
Hood Canal domains. The left vertical panel of **Fig. 6** shows mean seasonal cycles from the Cape
Elizabeth and Chá bă moorings and the Columbia River and Outer Coast regions. The two
moorings lie within the Outer Coast region and display similar SSS and SST seasonal cycles,
suggesting that large-scale rather than local physical processes dominate the seasonal cycles at
both locations. However, there are notable differences in the seasonal cycles of carbonate system
variables between moorings, which indicates that carbonate chemistry on the Washington coast is
somewhat heterogeneous and cannot be adequately characterized with observations from a single
mooring. In contrast to the moorings and Outer Coast domain, the Columbia River region exhibits
a SSS and TA depression during spring/early-summer in concert with the maximum in Columbia
River discharge. The fresh water plume is often advected southward due to seasonal winds and
does not greatly influence salinity in the neighboring Outer Coast region or at the Cape Elizabeth
mooring during the summer months.

Gray shading around the Cape Elizabeth mooring seasonal cycles reflects the average range of
variability about the monthly mean (**Eq. 2**; $\overline{m\sigma}$). Sensors on the Cape Elizabeth and Chá bă
moorings collect samples every 3 hours, which means that one year of data equates to ~3,000
observations, or ~250 observations per month (though there are occasional gaps in the time series;
**Fig. 1b**). With multiple years of observations (**Appendix B**), data from these moorings provide
robust constraints on the average range of sub-monthly variability. General agreement between the
mooring data and relatively poorly-sampled regions is therefore somewhat exceptional, as many
regional carbonate system values fall within the ±1σ envelope. This finding underscores that
seasonal variations outcompete the influence of vertical gradients associated with stratification
that would work to create discrepancies between the seasonal cycle data products.





**Figure 6.** Seasonal cycles of (**a-c**) SSS, (**d-f**) SST, (**g-i**) $f$CO₂, (**j-l**) DIC, (**m-o**) TA, (**p-r**) pH_T, (**s-u**) Ω_Ar, and (**v-x**) RF for regions and moorings located along the outer Washington Coast, in Puget





Sound, and in Hood Canal. Asterisks are used for the regions. Black lines with gray shading
highlight data from one mooring within each domain (see figure keys in subplots **a** to **c**). Gray
shading shows the average range of variability about the mean during each month (i.e., average
standard deviation across years derived from variance estimates). See also **Appendix D Figures**.

Moving inland, the middle vertical panel of **Fig. 6** shows the Carr Inlet and Point Wells
moorings and the Puget Sound region. The two moorings display similar SSS and TA seasonal
cycles in terms of magnitude and timing, and general agreement in SST with a larger amplitude
seasonal cycle at the Carr Inlet mooring due to more stratified conditions relative to the well-mixed
Point Wells site in the main basin of Puget Sound. The Puget Sound region generally follows the
mooring SSS and SST results, with some erratic variations likely caused by the much smaller
number of observations (**Appendix B**). Carbonate chemistry seasonal cycles from the Carr Inlet
mooring and Puget Sound region are similar in pattern but are offset due to the lower TA values
found throughout Puget Sound for nearly all months. Differences in the timing of seasonal
agreement between the carbon variables at Carr Inlet and Puget Sound may be attributable to
stratification and sampling depth; however, it is also important note that the carbonate variables
from Carr Inlet are based on only ~1.5 years of observations.

Finally, the right vertical panel of **Fig. 6** shows results from the Twanoh, Dabob Bay, and
Hoodsport moorings as well as the Hood Canal region. There are significant differences in SSS
across sites, while SST seasonal cycles are similar for Twanoh and Dabob Bay as well as for
Hoodsport and the Hood Canal region. Seasonal SSS variations at Twanoh and Hoodsport are
similar in timing and amplitude, but SSS is depressed at Twanoh due to its proximity to the
Skokomish River. This influence is muted at the Hoodsport mooring and even more so at the
Dabob Bay mooring (and correspondingly evident in TA), which is located much farther north and
outside of the main channel. The moorings, thus, depict a range of diverse physical conditions
within Hood Canal that depend on the sampling location. This is relevant because seasonal cycles
for the Hood Canal region were determined using far fewer discrete samples that were collected
across the entire domain (**Appendix B**), and thus blend all of the nuanced environments within the
canal. Though observations are not presently available to show the range of variability across the
Puget Sound region, this issue is likely also important when interpreting seasonal cycles from that
domain.




The seasonal cycles for carbonate system variables are unique across sites within Hood Canal

with somewhat better agreement found between the Twanoh and Dabob Bay moorings. The

pycnocline in Hood Canal is typically strong and often located at ~5-8 m. Thus, the 10 m depth-

integrated water samples used for the regional assessment would include colder, saltier, and more

carbon rich waters than the surface moorings. This influence can be seen in SSS and DIC where

the Hood Canal region displays higher values than the moorings during almost all months. This is

particularly notable for SSS since the mooring estimates come from averaging the top 3 m of

profiling mooring observations. Thus, the region and mooring SSS (and SST) discrepancies would

likely be larger if the mooring values actually came from the top 1 m of water, where the $f$CO$_2$

measurements are made. This is supported by the extremely high $f$CO$_2$ and low pH values observed

in the Hood Canal region during winter that significantly exceed what is found at the moorings.

Better agreement in $f$CO$_2$ and pH exists across sites during summer, and there are fewer

discrepancies for $\Omega_{Ar}$ and RF at all locations throughout the year due to the large dependence of

both parameters on DIC:TA, which is similar across sites.

Having discussed important nuances between the region and mooring seasonal cycles within

these domains, it is important to also consider the large-scale gradients in carbonate chemistry

across the domains. Taking a broader view of **Fig. 6**, greater environmental context is gained by

comparing results from the open North Pacific Ocean and the Salish Sea. The large decline in SSS

and increase in SST as waters become more stratified and heavily influenced by rivers when

moving toward land greatly outcompetes any small differences found among sites within a given

domain. There is an accompanying pattern of DIC decline along this path; however, TA exhibits

even larger declines, resulting in elevated DIC:TA and thus more poorly buffered waters (higher

RF values) within the Salish Sea. This leads to much higher $f$CO$_2$ values and lower pH and $\Omega_{Ar}$

values in Hood Canal and Puget Sound than on the outer Washington coast. Since the North Pacific

Ocean is the sole source of marine water to the Salish Sea, elevated carbon content within the

Salish Sea reflects longer retention times and potentially intensified carbon cycling due to local

nutrient and carbon dioxide pollution (Feely et al., 2010; Pelletier et al., 2017).




In addition to gradients in annual mean values of carbonate system parameters, there is a
significant increase in the amplitude of seasonal cycles and in the range of sub-monthly variability
moving landward. Unexpectedly, the seasonal evolution of sub-monthly variability is unique to
different domains. For example, at the Twanoh mooring $f$CO$_2$, pH, and $\Omega_{Ar}$ vary significantly more
about the mean value during winter than during summer; however, at Carr Inlet and Cape
Elizabeth, the opposite is true. This kind of insight, which emerges from sustained observing,
provides precisely the type of information necessary to streamline OA monitoring, adaption, and
research. For example, understanding how natural variability amplifies and attenuates throughout
the year could improve our interpretations of sparse data, since samples collected during seasons
with low sub-monthly variability may better represent the monthly mean condition. Additionally,
the season exhibiting the lowest range of variability for a given parameter will likely be first to
exit the envelope of natural variability as anthropogenic carbon accumulates (Hauri et al., 2013;
Sutton et al., 2016). This nuanced information may be useful for determining when organisms will
begin to experience persistent anomalous conditions and for designing experiments that accurately
reflect nonstationarity in natural variability throughout the year.

### 4.2. Important limitations to consider
Although the average seasonal cycles herein may help to inform OA monitoring and research
conducted throughout the region, there are important limitations that must be acknowledged. The
region-specific seasonal cycles are meant to provide large-scale, environmental context and an
understanding of how the carbonate system varies across the broader domain. The moorings fill in
temporal observing gaps and give robust statistical estimates of mean conditions at specific
locations, as well as the average variability around those means. General agreement among the
region and mooring seasonal cycles lends confidence to the results, but is not necessarily expected
due to data paucity coupled with natural temporal variability and spatial heterogeneity within the
various regions (e.g., **Fig. 6**). Opportunities for discrepancy were especially evident in the more
stratified domains, such as Hood Canal and Puget Sound, due to differences in sampling depth for
the various datasets. While seasonal forcing appears to dominate over stratification and vertical
gradient issues in all domains, prior work has clearly shown that surface conditions do not reflect
bottom water conditions (Feely et al., 2010; Pelletier et al., 2017). General agreement between





data products is likely due to all depths residing within the euphotic zone such that vertical
gradients are minimized relative to those found throughout the full water column.

A key challenge in developing seasonal cycle estimates for the smaller regional domains (e.g.,
Hood Canal and Puget Sound) is the dearth of data that renders these seasonal cycles more
susceptible to biases that may result from non-homogenous conditions (e.g., proximity to rivers),
interannual variability, or anomalous events (e.g., storms). As discrete observations amass, this
issue will attenuate, but at present is something that needs to be considered when using the
information provided herein. Additionally, regional seasonal cycles were evaluated using rigid
domain boundaries, which we know to be dynamic in reality. Thus, caution should be used when
relying on our estimates to evaluate samples collected near one of the boundaries. The use of a
static regional TA-salinity relationship also has important implications for how carbonate system
seasonal cycles can be interpreted. The TA-salinity relationship homogenizes the influence of
freshwater alkalinity input (organic and inorganic) from numerous rivers in the region. As a result,
the regression may not work perfectly near a specific river mouth. In these cases (e.g., the Twanoh
mooring), the TA estimates may be biased and the bias may change seasonally. Importantly, **Table**
**5** displays how these biases can propagate to other carbonate system variables. For all parameters,
these errors are significantly smaller than the seasonal cycle amplitudes and, in most cases, are
smaller than $\pm 1\sigma$ values from the moorings (**Appendix B**). This is also true of the *in situ*
measurement errors discussed throughout Sect. 2.

### *4.3. Understanding what the data do and do not reveal*
Characterizing average seasonal cycles throughout the study domain reveals important
insights about how different types of observations can be used to better understand the dynamic
coastal region. Discrete measurements made sporadically over space and time can be challenging
to interpret without a vast number of data points, which are both cost- and time-intensive to collect.
This limitation arises from spatial heterogeneity and the range of temporal scales over which
variability can occur in the coastal zone (e.g., < days to > decades). Discrete observations provide
a snapshot in time that may or may not represent the mean condition. Even with multiple years of
observations, and no interannual variability, the true mean can be elusive due to sub-monthly
variations. What discrete observations do extremely well is unveil the connectivity of spatial



domains by sampling across regions over a fixed period of time (e.g., research cruise). This too
has uncertainty resulting from environmental heterogeneity and chemical gradients that are
nonstationary over time, but is at present the most effective and efficient method of making dual-
parameter, high-quality carbon observations across a wide spatial area.

Moorings provide something altogether different. Sustained, high-frequency observations fill

in temporal uncertainties, integrating over (often) unknown spatial scales that vary with tidal
cycles, surface currents, and depth, allowing for robust determination of the mean condition and
the average variability about the mean within the represented domain. This type of information
lends context to discrete samples by providing bounds on the domain within which discrete
samples may be expected to reside. **Figure 7**, for example, shows the monthly mean $f\mathrm{CO}_2$ seasonal
cycle from the Cape Elizabeth mooring as well as the same data shifted upward by 50 μatm, which
is meant to represent an anomalous year, where the range of sub-monthly variability (shading) is
identical for each year. Two discrete $f\mathrm{CO}_2$ samples collected during one "normal" and one
"anomalous" June could span over 100 μatm, hampering determination of which sample best
reflects the mean conditions. Mooring observations collected over the same two years would give
an average seasonal cycle that falls in-between the two lines shown in **Fig. 7**, and the discrete
samples would thus lie outside of the associated ±1σ window. This would make it tempting to
conclude that the discrete samples are outliers when the data actually indicate that one of the
sampling years was anomalous. The true climatology would eventually emerge with additional
years of observations. Thus, firm conclusions about what is or isn't an anomaly or outlier may be
extremely challenging to decipher without a sustained time series and information about the
magnitude of sub-monthly variability.

To summarize, the array of time and space scales over which chemical variability occurs in

U.S. West Coast waters hinders our ability to adequately describe the coastal system without the
dual effort of ship-based and moored observations that provide complementary information. The
seasonal cycles presented herein are an example of what can be achieved from such efforts and
serve as a starting point from which strategic improvements to observational networks can be
made.



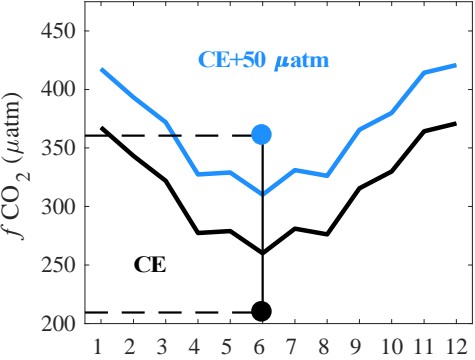

**Figure 7**. Schematic showing the Cape Elizabeth (CE) mooring monthly mean $f$CO₂ seasonal cycle
(black) and the seasonal cycle +50 µatm (blue), which is meant to reflect an anomalous year. Black
and blue shading are identical in magnitude and show the mean standard deviation ($\overline{m\sigma}$). The filled
circles show two hypothetical discrete samples collected during June of each year.

**5. Conclusions**
Quantifying modern, human-induced changes in surface ocean chemistry requires a baseline
from which to assess alterations. These baselines often do not exist in coastal regions where large-
magnitude and high-frequency natural oscillations can overwhelm the secular anthropogenic
signal. This inhibits our understanding of how coastal environments are changing and will continue
to do so until these domains are adequately characterized. In an effort to galvanize baseline
development for seawater parameters relevant to the study of ocean acidification, we have
compiled ~100,000 *in situ* carbonate system observations from marine surface waters near
Washington State to characterize carbonate system variability throughout the region. By merging
$f$CO₂ observations from the SOCAT-v4 database with dissolved inorganic carbon (DIC), total
alkalinity (TA), and pH$_T$ measurements made during discrete sampling efforts, we have estimated
the average seasonal cycles of numerous carbonate system parameters in marine surface waters
from eight distinct regions within the Pacific Northwest. Agreement between these regional
estimates and the average seasonal cycles determined at nine surface moorings throughout
Washington State coastal waters indicates a coherent, large-scale gradient in carbonate chemistry
from the open North Pacific Ocean into the Salish Sea that is seasonally variable. Near surface
salinity, DIC, TA, aragonite saturation state, and pH decline significantly along this path. Larger
declines in TA relative to DIC lead to elevated $f$CO₂ and Revelle Factor values within the Salish
Sea, indicating a lower capacity of these inland waters to absorb anthropogenic carbon than
offshore waters at the same latitude. These large-scale gradients persist throughout the year but





vary seasonally in magnitude with unique variables expressing larger gradients during different
seasons. For all carbonate system parameters, the seasonal cycle amplitudes and ranges of sub-
monthly variability are larger within the Salish Sea, indicating that organisms living in the Pacific
Northwest experience starkly different chemical environments across the study domain.

In addition to providing environmental context, this analysis reveals important insights about
the extreme care required to accurately interpret spatially and temporally nonuniform observations
from the coastal zone. Spatial heterogeneity was discovered within all regional domians evaluated,
indicating that chemical conditions, even in semi-enclosed domains, cannot be adequately
characterized using observations from a single mooring. Additionally, the wide range of sub-
monthly variability found at the mooring sites suggests that it may be difficult to determine what
is and isn't an outliner in a discrete dataset, particularly in the context of interannual variability.
By pairing multiple types of datasets (discrete, underway, and sustained time series) our ability to
interpret complex coastal environments can be enhanced by more than what is achievable from
any one observing approach. It is our intent that the insights gained from the seasonal cycle
characterizations herein help to guide strategic monitoring, management, and scientific inquiry
that leads to improved baseline development and water quality assessment in Washington State.

**6. Appendix A**
**Table A1**. Discrete datasets incorporated into the analysis. The observing period and the number
of phosphate, silicate, SSS, and SST observations for each cruise are provided. Data DOIs
matching those in Table 1 are also listed. Phosphate and silicate samples were analyzed following
standard protocols (UNESCO, 1994) during all cruises.

| NUTRIENTS | Obs. Period | Phosphate Obs. | Silicate Obs. | SSS Obs. | SST Obs. | Data DOI |
|---|---|---|---|---|---|---|
| SOCAT-v4[1] | 1970-2015 | 0 | 0 | 52,528 | 57,422 | https://doi.pangaea.de/10.1594/PANGAEA.866856 |
| NOAA WCOA | 05/2007 | 11 | 11 | 11 | 11 | 10.3334/CDIAC/otg.CLIVAR_NACP_West_Coast_Cruise_2007 |
| UW PRISM* | 02/2008 | 74 | 78 | 80 | 80 | https://doi.org/10.5281/zenodo.1184657 |
| UW PRISM/EPA* | 08/2008 | 105 | 105 | 122 | 122 | https://doi.org/10.5281/zenodo.1184657 |
| PacOOS* | 08/2009 | 8 | 8 | 8 | 8 | https://doi.org/10.5281/zenodo.1184657 |
| UW PRISM* | 09/2009 | 38 | 38 | 38 | 38 | https://doi.org/10.5281/zenodo.1184657 |
| PacOOS* | 05/2010 | 13 | 13 | 13 | 13 | https://doi.org/10.5281/zenodo.1184657 |
| PacOOS* | 08/2010 | 22 | 22 | 22 | 22 | https://doi.org/10.5281/zenodo.1184657 |
| UW PRISM* | 11/2010 | 39 | 39 | 40 | 40 | https://doi.org/10.5281/zenodo.1184657 |
| UW/Chá bǎ* | 05/2011 | 0 | 0 | 1 | 1 | https://doi.org/10.5281/zenodo.1184657 |
| NOAA WCOA | 08/2011 | 11 | 11 | 37 | 37 | 10.3334/CDIAC/OTG.COAST_WCOA2011 |





| | | | | | | |
|---|---|---|---|---|---|---|
| UW PRISM* | 10/2011 | 38 | 38 | 60 | 60 | https://doi.org/10.5281/zenodo.1184657 |
| UW/NANOOS* | 05/2012 | 0 | 0 | 4 | 4 | https://doi.org/10.5281/zenodo.1184657 |
| NOAA WCOA | 09/2012 | 40 | 43 | 43 | 43 | 10.3334/CDIAC/OTG.COAST_WCOA2012 |
| UW/Chá bă* | 01/2013 | 0 | 0 | 2 | 2 | https://doi.org/10.5281/zenodo.1184657 |
| UW/NANOOS* | 04/2013 | 1 | 1 | 19 | 19 | https://doi.org/10.5281/zenodo.1184657 |
| NOAA WCOA | 08/2013 | 59 | 59 | 59 | 59 | 10.7289/V5C53HXP |
| UW/NANOOS* | 09/2013 | 14 | 14 | 19 | 19 | https://doi.org/10.5281/zenodo.1184657 |
| UW/NANOOS* | 06/2014 | 10 | 10 | 10 | 10 | https://doi.org/10.5281/zenodo.1184657 |
| ECY | 06/2014 | 8 | 8 | 17 | 17 | https://doi.org/10.5281/zenodo.1184657 |
| WOAC* | 07/2014 | 61 | 61 | 68 | 68 | https://doi.org/10.5281/zenodo.1184657 |
| ECY | 07/2014 | 0 | 0 | 12 | 12 | https://doi.org/10.5281/zenodo.1184657 |
| ECY | 08/2014 | 0 | 0 | 18 | 18 | https://doi.org/10.5281/zenodo.1184657 |
| WOAC* | 09/2014 | 64 | 64 | 64 | 64 | https://doi.org/10.5281/zenodo.1184657 |
| ECY | 09/2014 | 0 | 0 | 14 | 14 | https://doi.org/10.5281/zenodo.1184657 |
| UW/Chá bă* | 10/2014 | 19 | 19 | 19 | 19 | https://doi.org/10.5281/zenodo.1184657 |
| ECY | 10/2014 | 0 | 0 | 9 | 9 | https://doi.org/10.5281/zenodo.1184657 |
| ECY | 11/2014 | 6 | 6 | 9 | 9 | https://doi.org/10.5281/zenodo.1184657 |
| ECY | 01/2015 | 13 | 12 | 13 | 13 | https://doi.org/10.5281/zenodo.1184657 |
| ECY | 02/2015 | 11 | 11 | 15 | 15 | https://doi.org/10.5281/zenodo.1184657 |
| ECY | 03/2015 | 8 | 8 | 20 | 20 | https://doi.org/10.5281/zenodo.1184657 |
| ECY | 04/2015 | 5 | 5 | 15 | 15 | https://doi.org/10.5281/zenodo.1184657 |
| ECY | 05/2015 | 2 | 2 | 6 | 6 | https://doi.org/10.5281/zenodo.1184657 |
| **TOTAL** | | **680** | **686** | **53,415** | **58,309** | |

[1]The coastal subset of SOCAT-v4 data can be easily accessed here: https://www.socat.info/index.php/version-4/
*Full dataset in preparation for submission to a long-term data repository.

**7.  Appendix B**
**Table B1**. Average monthly SSS seasonal cycles for the eight unique regions. The total number
of observations (# Obs.) contributing to the monthly average estimate is included.

| SSS | Jan. | Feb. | Mar. | Apr. | May | June | July | Aug. | Sept. | Oct. | Nov. | Dec. |
|---|---|---|---|---|---|---|---|---|---|---|---|---|
| North Pacific (NP) | 32.3 | 32.5 | 32.9 | 32.3 | 32.1 | 31.9 | 31.7 | 31.9 | 32.0 | 32.1 | 32.2 | 32.4 |
| # Obs. | 163 | 439 | 250 | 66 | 305 | 2791 | 4024 | 1812 | 504 | 88 | 109 | 29 |
| | | | | | | | | | | | | |
| British Columbia (BC) | 31.3 | 31.7 | 31.4 | 31.3 | 31.0 | 31.7 | 31.4 | 31.8 | 31.8 | 31.9 | 31.7 | 31.2 |
| # Obs. | 162 | 647 | 205 | 160 | 452 | 1107 | 8067 | 1735 | 250 | 229 | 110 | 100 |
| | | | | | | | | | | | | |
| Outer Coast (OC) | 32.1 | 32.2 | 32.2 | 31.7 | 31.5 | 31.2 | 31.0 | 31.8 | 31.8 | 31.8 | 32.3 | |
| # Obs. | 57 | 65 | 170 | 7 | 34 | 1700 | 6969 | 5585 | 749 | 5 | 13 | |
| | | | | | | | | | | | | |
| Columbia River (CR) | 32.1 | 31.6 | 30.7 | 29.1 | 29.1 | 30.1 | 30.0 | 29.9 | 30.7 | 31.4 | 32.3 | |
| # Obs. | 120 | 19 | 98 | 23 | 135 | 3456 | 1545 | 3059 | 719 | 24 | 46 | |
| | | | | | | | | | | | | |
| Juan de Fuca (JF) | 30.4 | 30.7 | 30.8 | 30.1 | 30.8 | 31.0 | 30.8 | 31.0 | 31.3 | 31.1 | 31.2 | 31.3 |
| # Obs. | 68 | 2099 | 172 | 48 | 274 | 627 | 2046 | 814 | 152 | 44 | 43 | 40 |
| | | | | | | | | | | | | |
| Strait of Georgia (SG) | | 28.7 | | 28.1 | 28.0 | 27.7 | 22.9 | 29.3 | 29.9 | 29.9 | | |
| # Obs. | | 192 | | 5 | 58 | 32 | 5 | 35 | 6 | 1 | | |
| | | | | | | | | | | | | |
| Hood Canal (HC) | 24.1 | 28.3 | 25.7 | 26.0 | 28.5 | 27.0 | 27.7 | 28.8 | 29.5 | 28.5 | 27.9 | |
| # Obs. | 1 | 24 | 3 | 3 | 3 | 3 | 27 | 28 | 6 | 71 | 25 | |



| | | | | | | | | | | | |
|---|---|---|---|---|---|---|---|---|---|---|---|
| Puget Sound (PS) | 21.8 | 28.1 | 23.9 | 25.2 | 29.3 | 27.8 | 27.8 | 28.8 | 29.0 | 29.1 | 27.4 |
| *# Obs.* | 12 | 57 | 17 | 12 | 35 | 93 | 287 | 257 | 32 | 49 | 15 |

**Table B2**. Average monthly SST seasonal cycles for the eight unique regions. The total number of observations (# Obs.) contributing to the monthly average estimate is included.

| SST (ºC) | Jan. | Feb. | Mar. | Apr. | May | June | July | Aug. | Sept. | Oct. | Nov. | Dec. |
|---|---|---|---|---|---|---|---|---|---|---|---|---|
| North Pacific (NP) | 9.3 | 9.1 | 9.7 | 10.1 | 11.6 | 13.5 | 15.5 | 16.7 | 16.1 | 14.9 | 12.2 | 9.6 |
| *# Obs.* | 163 | 446 | 250 | 66 | 364 | 2791 | 4026 | 1922 | 504 | 88 | 109 | 29 |
| British Columbia (BC) | 9.2 | 8.8 | 8.7 | 9.8 | 11.5 | 12.5 | 13.8 | 13.3 | 13.8 | 11.9 | 10.9 | 9.0 |
| *# Obs.* | 166 | 644 | 205 | 160 | 567 | 1102 | 8078 | 1938 | 250 | 229 | 110 | 100 |
| Outer Coast (OC) | 9.1 | 9.3 | 9.8 | 9.9 | 11.2 | 12.6 | 15.7 | 14.3 | 14.2 | 14.8 | 12.0 | |
| *# Obs.* | 57 | 65 | 170 | 7 | 34 | 1700 | 6969 | 6301 | 750 | 5 | 13 | |
| Columbia River (CR) | 9.7 | 8.3 | 10.5 | 10.1 | 12.2 | 12.9 | 15.1 | 15.5 | 15.9 | 15.3 | 12.3 | |
| *# Obs.* | 120 | 19 | 98 | 23 | 135 | 3456 | 1545 | 3614 | 719 | 24 | 46 | |
| Juan de Fuca (JF) | 8.1 | 7.6 | 8.5 | 9.4 | 10.0 | 10.3 | 11.1 | 10.9 | 11.2 | 10.9 | 10.2 | 8.1 |
| *# Obs.* | 71 | 2108 | 172 | 48 | 378 | 628 | 1953 | 816 | 153 | 44 | 43 | 39 |
| Strait of Georgia (SG) | | 7.2 | | 9.1 | 11.7 | 11.8 | 14.7 | 13.8 | 12.0 | 11.7 | | |
| *# Obs.* | | 197 | | 5 | 59 | 33 | 5 | 35 | 6 | 1 | | |
| Hood Canal (HC) | 9.2 | 7.8 | 10.1 | 10.9 | 12.0 | 13.9 | 16.5 | 14.2 | 13.9 | 11.6 | 10.8 | |
| *# Obs.* | 1 | 24 | 3 | 3 | 3 | 3 | 27 | 28 | 6 | 71 | 25 | |
| Puget Sound (PS) | 9.0 | 7.8 | 10.0 | 11.1 | 10.1 | 12.1 | 13.8 | 13.8 | 13.4 | 13.2 | 11.5 | |
| *# Obs.* | 12 | 57 | 17 | 12 | 35 | 93 | 298 | 257 | 32 | 49 | 15 | |

**Table B3**. Average monthly TA seasonal cycles for the eight unique regions. The total number of observations (# Obs.) contributing to the monthly average estimate is included.

| TA (µmol kg⁻¹) | Jan. | Feb. | Mar. | Apr. | May | June | July | Aug. | Sept. | Oct. | Nov. | Dec. |
|---|---|---|---|---|---|---|---|---|---|---|---|---|
| North Pacific (NP) | 2187 | 2194 | 2214 | 2185 | 2175 | 2165 | 2155 | 2167 | 2171 | 2176 | 2183 | 2189 |
| *# Obs.* | 163 | 439 | 250 | 66 | 305 | 2791 | 4024 | 1812 | 504 | 88 | 109 | 29 |
| British Columbia (BC) | 2138 | 2156 | 2141 | 2138 | 2124 | 2157 | 2144 | 2163 | 2163 | 2169 | 2158 | 2133 |
| *# Obs.* | 162 | 647 | 205 | 160 | 452 | 1107 | 8067 | 1735 | 250 | 229 | 110 | 100 |
| Outer Coast (OC) | 2173 | 2181 | 2179 | 2160 | 2150 | 2135 | 2123 | 2166 | 2160 | 2158 | 2184 | |
| *# Obs.* | 57 | 65 | 170 | 7 | 34 | 1700 | 6969 | 5584 | 749 | 5 | 13 | |
| Columbia River (CR) | 2179 | 2151 | 2111 | 2034 | 2049 | 2082 | 2076 | 2080 | 2113 | 2142 | 2187 | |
| *# Obs.* | 120 | 19 | 98 | 23 | 135 | 3456 | 1545 | 3059 | 719 | 24 | 46 | |
| Juan de Fuca (JF) | 2097 | 2111 | 2116 | 2081 | 2115 | 2125 | 2113 | 2125 | 2140 | 2126 | 2133 | 2137 |
| *# Obs.* | 68 | 2099 | 172 | 48 | 267 | 627 | 2046 | 816 | 152 | 44 | 43 | 40 |
| Strait of Georgia (SG) | | 2012 | | 1984 | 1980 | 1965 | 1842 | 2045 | 2071 | 2073 | | |
| *# Obs.* | | 192 | | 5 | 58 | 32 | 5 | 35 | 6 | 1 | | |
| Hood Canal (HC) | 1941 | 2007 | 1870 | 1972 | 2011 | 1935 | 1975 | 2033 | 2063 | 1999 | 1967 | |
| *# Obs.* | 1 | 24 | 3 | 3 | 3 | 3 | 27 | 28 | 6 | 71 | 25 | |
| Puget Sound (PS) | 1795 | 1986 | 1894 | 1919 | 2046 | 1956 | 1963 | 2020 | 2023 | 2035 | 1989 | |





| | | | | | | | | | | | | |
|---|---|---|---|---|---|---|---|---|---|---|---|---|
| *# Obs.* | 12 | 57 | 17 | 12 | 35 | 93 | 287 | 257 | 32 | 49 | 15 | |


**Table B4**. Average monthly DIC seasonal cycles for the eight unique regions. The total number
of observations (# Obs.) contributing to the monthly average estimate is included.

| DIC (µmol kg⁻¹) | Jan. | Feb. | Mar. | Apr. | May | June | July | Aug. | Sept. | Oct. | Nov. | Dec. |
|---|---|---|---|---|---|---|---|---|---|---|---|---|
| North Pacific (NP) | 2019 | 2027 | 2029 | 1992 | 1974 | 1954 | 1941 | 1940 | 1950 | 1964 | 1990 | 2017 |
| *# Obs.* | 163 | 446 | 250 | 66 | 305 | 2791 | 4024 | 1814 | 504 | 88 | 109 | 29 |
| British Columbia (BC) | 1998 | 2015 | 1992 | 1944 | 1903 | 1928 | 1935 | 1959 | 1967 | 2012 | 2001 | 2000 |
| *# Obs.* | 162 | 627 | 205 | 160 | 462 | 1104 | 8067 | 1732 | 250 | 229 | 110 | 100 |
| Outer Coast (OC) | 2008 | 2023 | 2006 | 1941 | 1962 | 1918 | 1879 | 1928 | 1919 | 1957 | 1993 | |
| *# Obs.* | 57 | 65 | 170 | 7 | 34 | 1700 | 6969 | 5585 | 749 | 5 | 13 | |
| Columbia River (CR) | 2014 | 1988 | 1926.5 | 1852 | 1855 | 1874 | 1831 | 1841 | 1897 | 1938 | 1989 | |
| *# Obs.* | 120 | 19 | 98 | 23 | 135 | 3456 | 1545 | 3059 | 719 | 24 | 46 | |
| Juan de Fuca (JF) | 1993 | 2034 | 2020 | 1956 | 1970 | 2024 | 2016 | 2038 | 2056 | 2054 | 2031 | 2075 |
| *# Obs.* | 68 | 2103 | 172 | 48 | 301 | 633 | 2044 | 827 | 152 | 44 | 43 | 40 |
| Strait of Georgia (SG) | | 1966 | | 1857 | 1808 | 1833 | 1702 | 1905 | 1981 | 1958 | | |
| *# Obs.* | | 192 | | 5 | 58 | 32 | 5 | 35 | 6 | 1 | | |
| Hood Canal (HC) | 1921 | 1964 | 1628 | 1756 | 1822 | 1784 | 1783 | 1888 | 1944 | 1951 | 1972 | |
| *# Obs.* | 1 | 24 | 3 | 3 | 3 | 3 | 27 | 28 | 6 | 71 | 25 | |
| Puget Sound (PS) | 1778 | 1944 | 1844 | 1751 | 1915 | 1830 | 1820 | 1881 | 1919 | 1932 | 1959 | |
| *# Obs.* | 12 | 57 | 17 | 12 | 35 | 93 | 287 | 257 | 32 | 49 | 15 | |


**Table B5**. Average monthly *f*CO₂ seasonal cycles for the eight unique regions. The total number
of observations (# Obs.) contributing to the monthly average estimate is included.

| $f\mathrm{CO_2}$ (µatm) | Jan. | Feb. | Mar. | Apr. | May | June | July | Aug. | Sept. | Oct. | Nov. | Dec. |
|---|---|---|---|---|---|---|---|---|---|---|---|---|
| North Pacific (NP) | 361 | 363 | 343 | 320 | 320 | 322 | 340 | 338 | 344 | 344 | 349 | 357 |
| *# Obs.* | 163 | 446 | 250 | 66 | 364 | 2791 | 4026 | 1876 | 504 | 88 | 109 | 29 |
| British Columbia (BC) | 402 | 413 | 380 | 300 | 261 | 290 | 335 | 349 | 368 | 445 | 404 | 432 |
| *# Obs.* | 164 | 648 | 205 | 160 | 562 | 1083 | 8078 | 1922 | 248 | 229 | 110 | 100 |
| Outer Coast (OC) | 358 | 385 | 356 | 264 | 337 | 288 | 282 | 275 | 285 | 359 | 351 | |
| *# Obs.* | 57 | 65 | 170 | 7 | 34 | 1700 | 6969 | 6301 | 750 | 5 | 13 | |
| Columbia River (CR) | 373 | 341 | 311 | 277 | 287 | 287 | 257 | 275 | 323 | 350 | 340 | |
| *# Obs.* | 120 | 19 | 98 | 23 | 135 | 3456 | 1545 | 3614 | 719 | 24 | 46 | |
| Juan de Fuca (JF) | 486 | 621 | 550 | 438 | 433 | 584 | 624 | 655 | 699 | 769 | 609 | 724 |
| *# Obs.* | 69 | 2110 | 172 | 48 | 367 | 638 | 2094 | 832 | 157 | 44 | 43 | 40 |
| Strait of Georgia (SG) | | 694 | | 386 | 304 | 400 | 300 | 466 | 621 | 500 | | |
| *# Obs.* | | 197 | | 5 | 59 | 33 | 5 | 35 | 6 | 1 | | |
| Hood Canal (HC) | 879 | 750 | 161 | 211 | 274 | 356 | 325 | 462 | 607 | 1032 | 1311 | |
| *# Obs.* | 1 | 24 | 3 | 3 | 3 | 3 | 27 | 28 | 6 | 71 | 25 | |
| Puget Sound (PS) | 794 | 742 | 654 | 283 | 286 | 416 | 422 | 487 | 614 | 607 | 948 | |
| *# Obs.* | 12 | 57 | 17 | 12 | 35 | 93 | 298 | 257 | 32 | 49 | 15 | |




**Table B6**. Average monthly pH seasonal cycles for the eight unique regions. The total number of observations (# Obs.) contributing to the monthly average estimate is included.


| pH | Jan. | Feb. | Mar. | Apr. | May | June | July | Aug. | Sept. | Oct. | Nov. | Dec. |
|---|---|---|---|---|---|---|---|---|---|---|---|---|
| North Pacific (NP) | 8.07 | 8.07 | 8.09 | 8.12 | 8.12 | 8.11 | 8.09 | 8.10 | 8.09 | 8.09 | 8.08 | 8.07 |
| *# Obs.* | 163 | 446 | 250 | 66 | 358 | 2791 | 4024 | 1770 | 504 | 88 | 109 | 29 |
| British Columbia (BC) | 8.02 | 8.02 | 8.05 | 8.14 | 8.18 | 8.16 | 8.11 | 8.09 | 8.07 | 8.00 | 8.03 | 8.00 |
| *# Obs.* | 160 | 647 | 205 | 160 | 455 | 1083 | 8067 | 1744 | 248 | 229 | 110 | 100 |
| Outer Coast (OC) | 8.07 | 8.05 | 8.07 | 8.19 | 8.10 | 8.15 | 8.16 | 8.14 | 8.16 | 8.07 | 8.08 | |
| *# Obs.* | 57 | 65 | 170 | 7 | 34 | 1700 | 6969 | 5585 | 749 | 5 | 13 | |
| Columbia River (CR) | 8.06 | 8.09 | 8.12 | 8.16 | 8.14 | 8.15 | 8.20 | 8.16 | 8.10 | 8.08 | 8.09 | |
| *# Obs.* | 120 | 19 | 98 | 23 | 135 | 3456 | 1545 | 3059 | 719 | 24 | 46 | |
| Juan de Fuca (JF) | 7.95 | 7.85 | 7.90 | 7.99 | 8.02 | 7.89 | 7.86 | 7.83 | 7.81 | 7.78 | 7.88 | 7.79 |
| *# Obs.* | 67 | 2110 | 172 | 48 | 301 | 635 | 2044 | 799 | 152 | 44 | 43 | 40 |
| Strait of Georgia (SG) | | 7.78 | | 8.03 | 8.13 | 8.02 | 8.08 | 7.98 | 7.85 | 7.93 | | |
| *# Obs.* | | 192 | | 5 | 58 | 32 | 5 | 35 | 6 | 1 | | |
| Hood Canal (HC) | 7.69 | 7.76 | 8.37 | 8.27 | 8.15 | 8.05 | 8.10 | 7.98 | 7.90 | 7.71 | 7.54 | |
| *# Obs.* | 1 | 24 | 3 | 3 | 3 | 3 | 27 | 28 | 6 | 71 | 25 | |
| Puget Sound (PS) | 7.71 | 7.76 | 7.81 | 8.16 | 7.99 | 8.00 | 8.02 | 7.97 | 7.88 | 7.87 | 7.67 | |
| *# Obs.* | 12 | 57 | 17 | 12 | 35 | 93 | 287 | 257 | 32 | 49 | 15 | |


**Table B7**. Average monthly $\Omega_{Ar}$ seasonal cycles for the eight unique regions. The total number of observations (# Obs.) contributing to the monthly average estimate is included.


| $\Omega_{Ar}$ | Jan. | Feb. | Mar. | Apr. | May | June | July | Aug. | Sept. | Oct. | Nov. | Dec. |
|---|---|---|---|---|---|---|---|---|---|---|---|---|
| North Pacific (NP) | 1.9 | 1.9 | 2.0 | 2.1 | 2.2 | 2.3 | 2.4 | 2.5 | 2.5 | 2.4 | 2.1 | 1.9 |
| *# Obs.* | 163 | 446 | 250 | 66 | 338 | 2791 | 4024 | 1814 | 504 | 88 | 109 | 29 |
| British Columbia (BC) | 1.6 | 1.6 | 1.7 | 2.1 | 2.4 | 2.5 | 2.3 | 2.3 | 2.2 | 1.8 | 1.8 | 1.6 |
| *# Obs.* | 161 | 647 | 205 | 160 | 455 | 1105 | 8067 | 1732 | 248 | 229 | 110 | 100 |
| Outer Coast (OC) | 1.9 | 1.8 | 1.9 | 2.4 | 2.1 | 2.4 | 2.7 | 2.6 | 2.6 | 2.2 | 2.1 | |
| *# Obs.* | 57 | 65 | 170 | 7 | 34 | 1700 | 6969 | 5585 | 749 | 5 | 13 | |
| Columbia River (CR) | 1.9 | 1.8 | 2.0 | 2.0 | 2.1 | 2.3 | 2.7 | 2.6 | 2.4 | 2.3 | 2.2 | |
| *# Obs.* | 120 | 19 | 98 | 23 | 135 | 3456 | 1545 | 3059 | 719 | 24 | 46 | |
| Juan de Fuca (JF) | 1.3 | 1.1 | 1.2 | 1.5 | 1.7 | 1.3 | 1.3 | 1.2 | 1.2 | 1.1 | 1.3 | 1.0 |
| *# Obs.* | 68 | 2110 | 172 | 48 | 300 | 633 | 2043 | 788 | 150 | 44 | 43 | 39 |
| Strait of Georgia (SG) | | 0.8 | | 1.5 | 1.9 | 1.6 | 1.6 | 1.7 | 1.2 | 1.4 | | |
| *# Obs.* | | 192 | | 5 | 58 | 32 | 5 | 35 | 6 | 1 | | |
| Hood Canal (HC) | 0.6 | 0.8 | 2.6 | 2.4 | 2.1 | 1.7 | 2.2 | 1.7 | 1.5 | 1.0 | 0.5 | |
| *# Obs.* | 1 | 24 | 3 | 3 | 3 | 3 | 27 | 28 | 6 | 71 | 25 | |
| Puget Sound (PS) | 0.6 | 0.8 | 0.8 | 1.9 | 1.5 | 1.5 | 1.7 | 1.6 | 1.3 | 1.3 | 0.7 | |
| *# Obs.* | 12 | 57 | 17 | 12 | 35 | 93 | 287 | 257 | 32 | 49 | 15 | |




**Table B8**. Average monthly Revelle Factor seasonal cycles for the eight unique regions. The total
number of observations (# Obs.) contributing to the monthly average estimate is included.

| Revelle Factor | Jan. | Feb. | Mar. | Apr. | May | June | July | Aug. | Sept. | Oct. | Nov. | Dec. |
|---|---|---|---|---|---|---|---|---|---|---|---|---|
| North Pacific (NP) | 13.0 | 13.1 | 12.5 | 12.1 | 11.8 | 11.5 | 11.3 | 11.0 | 11.1 | 11.4 | 12.1 | 12.9 |
| *# Obs.* | 163 | 446 | 250 | 66 | 356 | 2791 | 4024 | 1814 | 504 | 88 | 109 | 29 |
| British Columbia (BC) | 14.1 | 14.1 | 13.7 | 12.1 | 11.2 | 11.2 | 11.6 | 11.9 | 12.0 | 13.5 | 13.4 | 14.4 |
| *# Obs.* | 160 | 647 | 205 | 160 | 459 | 1082 | 8067 | 1727 | 248 | 229 | 110 | 100 |
| Outer Coast (OC) | 13.1 | 13.4 | 12.9 | 11.3 | 12.3 | 11.3 | 10.6 | 10.8 | 10.7 | 11.7 | 12.2 | |
| *# Obs.* | 57 | 65 | 170 | 7 | 34 | 1700 | 6969 | 5585 | 749 | 5 | 13 | |
| Columbia River (CR) | 13.1 | 13.2 | 12.2 | 12.2 | 11.8 | 11.5 | 10.5 | 10.7 | 11.2 | 11.6 | 11.9 | |
| *# Obs.* | 120 | 19 | 98 | 23 | 135 | 3456 | 1545 | 3059 | 719 | 24 | 46 | |
| Juan de Fuca (JF) | 15.7 | 17.2 | 16.2 | 14.7 | 14.0 | 15.9 | 16.1 | 16.5 | 16.6 | 17.1 | 15.8 | 17.8 |
| *# Obs.* | 67 | 2110 | 172 | 48 | 300 | 635 | 2043 | 795 | 152 | 44 | 43 | 39 |
| Strait of Georgia (SG) | | 18.6 | | 14.6 | 12.5 | 14.1 | 13.2 | 13.9 | 16.2 | 15.0 | | |
| *# Obs.* | | 192 | | 5 | 58 | 32 | 5 | 35 | 6 | 1 | | |
| Hood Canal (HC) | 19.9 | 18.7 | 10.3 | 11.2 | 11.8 | 13.0 | 11.7 | 13.7 | 14.9 | 16.3 | 18.5 | |
| *# Obs.* | 1 | 24 | 3 | 3 | 3 | 3 | 27 | 28 | 6 | 71 | 25 | |
| Puget Sound (PS) | 20.2 | 18.8 | 18.7 | 12.9 | 14.2 | 14.2 | 13.6 | 14.0 | 15.6 | 15.7 | 18.9 | |
| *Obs.* | 12 | 57 | 17 | 12 | 35 | 93 | 287 | 257 | 32 | 49 | 15 | |


## 8.  Appendix C

**Table C1**. Average monthly SSS seasonal cycles for the nine Washington State moorings. The
average monthly standard deviation (StDev) and total number of observations (# Obs.)
contributing to the monthly average estimate are included.

| SSS | Jan. | Feb. | Mar. | Apr. | May | June | July | Aug. | Sept. | Oct. | Nov. | Dec. |
|---|---|---|---|---|---|---|---|---|---|---|---|---|
| Chá bă (CB) | 30.3 | 30.7 | 30.6 | 30.5 | 30.8 | 31.2 | 31.5 | 31.9 | 31.9 | 31.8 | 31.8 | 31.4 |
| *StDev* | 1.2 | 1.2 | 1.6 | 1.1 | 0.4 | 1.2 | 0.8 | 0.3 | 0.5 | 0.5 | 0.5 | 0.7 |
| *# Obs.* | 398 | 224 | 248 | 280 | 485 | 1,080 | 1,461 | 1,648 | 1,346 | 1,029 | 646 | 496 |
| Cape Elizabeth (CE) | 31.1 | 30.7 | 30.9 | 30.4 | 30.5 | 31.2 | 31.7 | 32.0 | 31.9 | 31.8 | 31.8 | 31.5 |
| *StDev* | 1.3 | 1.5 | 1.4 | 1.3 | 1.5 | 1.1 | 0.7 | 0.3 | 0.5 | 0.5 | 0.6 | 0.8 |
| *# Obs.* | 1,488 | 1,352 | 1,295 | 1,199 | 1,161 | 1,246 | 1,715 | 1,812 | 1,621 | 1,756 | 1,538 | 1,484 |
| Twanoh (TW) | 24.5 | 24.5 | 24.7 | 24.6 | 24.7 | 25.1 | 25.7 | 26.4 | 26.5 | 26.2 | 25.8 | 25.1 |
| *StDev* | 2.7 | 2.3 | 2.3 | 1.8 | 1.5 | 1.0 | 1.1 | 0.7 | 1.0 | 1.7 | 2.5 | 2.7 |
| *# Obs.* | 684 | 913 | 1,443 | 2,678 | 3,332 | 3,234 | 3,169 | 3,664 | 3,993 | 3,220 | 1,545 | 591 |
| Dabob Bay (DB) | 29.2 | 28.3 | 27.6 | 26.8 | 26.6 | 26.6 | 26.2 | 27.3 | 28.3 | 28.9 | 28.0 | 28.6 |
| *StDev* | 0.1 | 0.6 | 0.8 | 0.5 | 0.5 | 0.8 | 1.0 | 0.9 | 0.3 | 0.5 | 1.0 | 0.9 |
| *# Obs.* | 34 | 33 | 85 | 109 | 174 | 286 | 767 | 1,167 | 1,138 | 667 | 163 | 39 |
| Carr Inlet (CI) | 28.8 | 28.5 | 28.2 | 28.1 | 28.2 | 28.4 | 28.8 | 29.1 | 29.4 | 29.7 | 29.7 | 29.2 |
| *StDev* | 0.2 | 0.2 | 0.2 | 0.2 | 0.2 | 0.1 | 0.1 | 0.1 | 0.1 | 0.2 | 0.1 | 0.2 |
| *# Obs.* | 349 | 347 | 1,008 | 937 | 820 | 1,161 | 1,629 | 1,641 | 1,611 | 1,566 | 716 | 584 |
| Bellingham Bay (BB) | 28.5 | 19.3 | 24.3 | 23.0 | 24.4 | 25.4 | 24.4 | 25.9 | 27.3 | 24.4 | 22.0 | 26.9 |



|  |  | Jan. | Feb. | Mar. | Apr. | May | June | July | Aug. | Sept. | Oct. | Nov. | Dec. |
|---|---|---|---|---|---|---|---|---|---|---|---|---|---|
|  | *StDev* | 2.9 | 8.5 | 5.1 | 4.9 | 3.3 | 3.3 | 2.9 | 2.2 | 2.6 | 6.7 | 7.7 | 3.9 |
|  | *# Obs.* | 844 | 2,102 | 4,308 | 3,611 | 4,365 | 4,227 | 4,317 | 3,958 | 4,223 | 4,383 | 4,218 | 4,339 |
| Hoodsport (HS) |  | 25.8 | 25.9 | 26.2 | 26.4 | 25.9 | 26.4 | 26.4 | 27.0 | 27.5 | 27.7 | 27.4 | 26.3 |
|  | *StDev* | 2.3 | 2.2 | 1.9 | 1.4 | 1.4 | 1.2 | 1.2 | 0.9 | 1.0 | 1.4 | 1.9 | 2.1 |
|  | *# Obs.* | 346 | 309 | 593 | 1,325 | 1,991 | 1,477 | 2,669 | 3,241 | 2,332 | 1,682 | 850 | 371 |
| Point Wells (PW) |  | 28.3 | 27.8 | 28.2 | 28.1 | 28.4 | 28.2 | 28.8 | 29.4 | 29.8 | 30.0 | 29.3 | 28.6 |
|  | *StDev* | 1.2 | 0.9 | 0.5 | 0.5 | 0.4 | 0.5 | 0.4 | 0.2 | 0.2 | 0.2 | 0.8 | 0.6 |
|  | *# Obs.* | 128 | 84 | 208 | 275 | 474 | 379 | 631 | 494 | 398 | 751 | 364 | 129 |
| North Buoy (NB) |  | 28.9 | 28.9 | 29.1 | 29.2 | 29.1 | 29.2 | 29.6 | 29.9 | 30.0 | 30.3 | 29.9 | 29.4 |
|  | *StDev* | 0.6 | 0.5 | 0.4 | 0.2 | 0.3 | 0.3 | 0.3 | 0.2 | 0.2 | 0.3 | 0.5 | 0.5 |
|  | *# Obs.* | 329 | 474 | 349 | 578 | 884 | 972 | 1,476 | 1,734 | 858 | 1,320 | 693 | 532 |


**Table C2**. Average monthly SST seasonal cycles for the nine Washington State moorings. The
average monthly standard deviation (StDev) and total number of observations (# Obs.)
contributing to the monthly average estimate are included.

| SST (ºC) |  | Jan. | Feb. | Mar. | Apr. | May | June | July | Aug. | Sept. | Oct. | Nov. | Dec. |
|---|---|---|---|---|---|---|---|---|---|---|---|---|---|
| Chá bă (CB) |  | 8.6 | 7.8 | 8.9 | 10.0 | 11.2 | 13.1 | 13.7 | 14.0 | 13.9 | 13.5 | 11.7 | 9.5 |
|  | *StDev* | 0.6 | 0.4 | 0.6 | 0.5 | 0.6 | 0.8 | 1.0 | 0.9 | 0.9 | 0.5 | 0.7 | 0.6 |
|  | *# Obs.* | 398 | 224 | 248 | 280 | 485 | 1,080 | 1,461 | 1,673 | 1,553 | 1,173 | 646 | 496 |
| Cape Elizabeth (CE) |  | 9.2 | 9.0 | 9.2 | 9.7 | 11.2 | 12.8 | 13.7 | 13.8 | 14.1 | 12.8 | 11.5 | 10.1 |
|  | *StDev* | 0.6 | 0.4 | 0.4 | 0.5 | 1.0 | 1.0 | 1.2 | 0.9 | 0.9 | 0.8 | 0.7 | 0.6 |
|  | *# Obs.* | 1,488 | 1,352 | 1,295 | 1,199 | 1,161 | 1,246 | 1,715 | 1,812 | 1,621 | 1,829 | 1,778 | 1,513 |
| Twanoh (TW) |  | 8.6 | 8.8 | 9.6 | 11.2 | 13.8 | 16.1 | 18.4 | 18.4 | 16.0 | 12.5 | 10.2 | 8.9 |
|  | *StDev* | 0.8 | 0.7 | 0.6 | 1.0 | 1.4 | 1.6 | 1.8 | 1.6 | 1.8 | 1.3 | 0.8 | 0.9 |
|  | *# Obs.* | 684 | 913 | 1,443 | 2,678 | 3,332 | 3,234 | 3,169 | 3,664 | 3,993 | 3,220 | 1,545 | 591 |
| Dabob Bay (DB) |  | 7.8 | 7.5 | 8.5 | 9.9 | 13.7 | 15.9 | 18.4 | 18.4 | 16.0 | 12.5 | 10.3 | 8.4 |
|  | *StDev* | 0.2 | 0.3 | 0.4 | 0.5 | 0.9 | 1.5 | 1.8 | 1.7 | 1.5 | 1.1 | 0.6 | 0.7 |
|  | *# Obs.* | 34 | 33 | 85 | 109 | 174 | 286 | 767 | 1,167 | 1,138 | 667 | 163 | 39 |
| Carr Inlet (CI) |  | 8.7 | 8.5 | 8.7 | 10.0 | 11.8 | 13.3 | 14.7 | 15.4 | 14.7 | 13.2 | 11.8 | 9.8 |
|  | *StDev* | 0.2 | 0.2 | 0.4 | 0.7 | 1.0 | 1.4 | 1.3 | 1.3 | 0.9 | 0.5 | 0.5 | 0.4 |
|  | *# Obs.* | 349 | 347 | 1,008 | 937 | 820 | 1,161 | 1,629 | 1,641 | 1,611 | 1,566 | 716 | 584 |
| Bellingham Bay (BB) |  | 6.7 | 8.2 | 9.1 | 12.2 | 14.3 | 14.9 | 17.4 | 17.6 | 14.7 | 11.4 | 9.9 | 7.4 |
|  | *StDev* | 0.7 | 0.7 | 0.6 | 1.2 | 1.2 | 1.2 | 1.6 | 1.3 | 1.3 | 0.9 | 1.0 | 1.2 |
|  | *# Obs.* | 844 | 2,102 | 4,308 | 3,611 | 4,365 | 4,227 | 4,317 | 3,958 | 4,223 | 4,383 | 4,218 | 4,339 |
| Hoodsport (HS) |  | 8.7 | 8.6 | 9.3 | 10.5 | 12.9 | 14.4 | 17.0 | 16.7 | 14.4 | 11.7 | 10.4 | 9.3 |
|  | *StDev* | 0.9 | 0.7 | 0.4 | 0.9 | 1.5 | 1.5 | 2.0 | 1.7 | 1.7 | 1.3 | 0.7 | 0.8 |
|  | *# Obs.* | 346 | 309 | 593 | 1,325 | 1,991 | 1,477 | 2,669 | 3,241 | 2,332 | 1,682 | 850 | 371 |
| Point Wells (PW) |  | 8.1 | 8.2 | 8.5 | 9.6 | 10.9 | 12.1 | 13.2 | 13.6 | 13.1 | 12.6 | 11.3 | 9.3 |
|  | *StDev* | 0.3 | 0.3 | 0.2 | 0.3 | 0.6 | 0.7 | 0.6 | 0.5 | 0.3 | 0.3 | 0.6 | 0.4 |
|  | *# Obs.* | 128 | 84 | 208 | 275 | 474 | 379 | 631 | 494 | 398 | 751 | 364 | 129 |
| North Buoy (NB) |  | 8.4 | 8.2 | 8.5 | 9.1 | 10.6 | 11.7 | 13.0 | 13.1 | 12.7 | 11.2 | 10.3 | 8.8 |
|  | *StDev* | 0.2 | 0.1 | 0.3 | 0.4 | 0.5 | 0.7 | 0.7 | 0.6 | 0.4 | 0.3 | 0.4 | 0.4 |
|  | *# Obs.* | 329 | 474 | 349 | 578 | 884 | 972 | 1,476 | 1,734 | 858 | 1,320 | 693 | 532 |






**Table C3**. Average monthly TA seasonal cycles for the eight Washington State moorings with
appropriate SSS observations (vast majority of SSS >20) to estimate TA. The average monthly
standard deviation (StDev) and total number of observations (# Obs.) contributing to the monthly
average estimate are included.

| TA (µmol kg⁻¹) | Jan. | Feb. | Mar. | Apr. | May | June | July | Aug. | Sept. | Oct. | Nov. | Dec. |
|---|---|---|---|---|---|---|---|---|---|---|---|---|
| Chá bă (CB) | 2,090 | 2,108 | 2,105 | 2,102 | 2,114 | 2,132 | 2,148 | 2,165 | 2,167 | 2,161 | 2,162 | 2,142 |
| *StDev* | 59 | 58 | 77 | 54 | 20 | 55 | 40 | 12 | 25 | 22 | 22 | 34 |
| *# Obs.* | 398 | 224 | 248 | 280 | 485 | 1,080 | 1,461 | 1,648 | 1,346 | 1,029 | 646 | 496 |
| Cape Elizabeth (CE) | 2,130 | 2,110 | 2,118 | 2,094 | 2,101 | 2,135 | 2,156 | 2,173 | 2,168 | 2,162 | 2,164 | 2,149 |
| *StDev* | 62 | 73 | 67 | 60 | 73 | 53 | 32 | 14 | 24 | 24 | 28 | 38 |
| *# Obs.* | 1,488 | 1,352 | 1,295 | 1,199 | 1,161 | 1,246 | 1,715 | 1,812 | 1,621 | 1,756 | 1,538 | 1,484 |
| Twanoh (TW) | 1,815 | 1,815 | 1,823 | 1,820 | 1,825 | 1,844 | 1,870 | 1,903 | 1,909 | 1,896 | 1,876 | 1,842 |
| *StDev* | 127 | 109 | 108 | 85 | 71 | 49 | 51 | 34 | 49 | 82 | 119 | 127 |
| *# Obs.* | 684 | 913 | 1,443 | 2,678 | 3,332 | 3,234 | 3,169 | 3,664 | 3,993 | 3,220 | 1,545 | 591 |
| Dabob Bay (DB) | 2,037 | 1,996 | 1,964 | 1,922 | 1,915 | 1,916 | 1,895 | 1,945 | 1,997 | 2,022 | 1,980 | 2,012 |
| *StDev* | 6 | 29 | 39 | 24 | 24 | 39 | 49 | 41 | 16 | 22 | 47 | 45 |
| *# Obs.* | 34 | 33 | 85 | 109 | 174 | 286 | 767 | 1,167 | 1,138 | 667 | 163 | 39 |
| Carr Inlet (CI) | 2,019 | 2,007 | 1,992 | 1,988 | 1,991 | 2,002 | 2,020 | 2,033 | 2,047 | 2,060 | 2,064 | 2,036 |
| *StDev* | 11 | 8 | 11 | 7 | 7 | 7 | 7 | 5 | 7 | 8 | 7 | 11 |
| *# Obs.* | 349 | 347 | 1,008 | 937 | 820 | 1,161 | 1,629 | 1,641 | 1,611 | 1,566 | 716 | 584 |
| Hoodsport (HS) | 1,890 | 1,891 | 1,897 | 1,904 | 1,881 | 1,904 | 1,904 | 1,932 | 1,957 | 1,968 | 1,959 | 1,907 |
| *StDev* | 93 | 86 | 82 | 67 | 69 | 55 | 55 | 44 | 48 | 65 | 80 | 91 |
| *# Obs.* | 337 | 302 | 590 | 1,323 | 1,991 | 1,477 | 2,669 | 3,241 | 2,332 | 1,681 | 843 | 361 |
| Point Wells (PW) | 1,995 | 1,972 | 1,992 | 1,986 | 1,998 | 1,989 | 2,020 | 2,050 | 2,066 | 2,078 | 2,042 | 2,010 |
| *StDev* | 58 | 41 | 25 | 22 | 20 | 25 | 19 | 12 | 7 | 11 | 40 | 28 |
| *# Obs.* | 128 | 84 | 208 | 275 | 474 | 379 | 631 | 494 | 398 | 751 | 364 | 129 |
| North Buoy (NB) | 2,026 | 2,023 | 2,033 | 2,038 | 2,036 | 2,038 | 2,055 | 2,073 | 2,076 | 2,090 | 2,072 | 2,047 |
| *StDev* | 29 | 22 | 19 | 12 | 15 | 16 | 15 | 12 | 12 | 15 | 22 | 25 |
| *# Obs.* | 329 | 474 | 349 | 578 | 884 | 972 | 1,476 | 1,734 | 858 | 1,320 | 693 | 532 |

**Table C4**. Average monthly DIC seasonal cycles for the five Washington State moorings with
sufficient observations to estimate DIC. The average monthly standard deviation (StDev) and total
number of observations (# Obs.) contributing to the monthly average estimate are included.

| DIC (µmol kg⁻¹) | Jan. | Feb. | Mar. | Apr. | May | June | July | Aug. | Sept. | Oct. | Nov. | Dec. |
|---|---|---|---|---|---|---|---|---|---|---|---|---|
| Chá bă (CB) | 1,936 | 1,964 | 1,973 | 1,882 | 1,846 | 1,850 | 1,891 | 1,922 | 1,950 | 1,960 | 1,981 | 1,975 |
| *StDev* | 47 | 54 | 72 | 56 | 34 | 66 | 57 | 41 | 36 | 25 | 29 | 33 |
| *# Obs.* | 398 | 221 | 245 | 279 | 484 | 1,080 | 1,461 | 1,648 | 1,346 | 1,028 | 645 | 495 |
| Cape Elizabeth (CE) | 1,975 | 1,949 | 1,944 | 1,898 | 1,891 | 1,893 | 1,915 | 1,925 | 1,944 | 1,958 | 1,986 | 1,987 |
| *StDev* | 54 | 70 | 64 | 57 | 71 | 54 | 47 | 41 | 49 | 41 | 34 | 34 |
| *# Obs.* | 1,485 | 1,347 | 1,295 | 1,199 | 1,158 | 1,244 | 1,715 | 1,812 | 1,619 | 1,753 | 1,506 | 1,484 |
| Twanoh (TW) | 1,783 | 1,722 | 1,630 | 1,644 | 1,688 | 1,705 | 1,709 | 1,726 | 1,784 | 1,747 | 1,779 | 1,815 |
| *StDev* | 67 | 97 | 88 | 80 | 73 | 56 | 50 | 33 | 54 | 81 | 104 | 75 |
| *# Obs.* | 371 | 392 | 506 | 725 | 875 | 877 | 743 | 961 | 1,650 | 1,145 | 613 | 363 |



| | Jan. | Feb. | Mar. | Apr. | May | June | July | Aug. | Sept. | Oct. | Nov. | Dec. |
|---|---|---|---|---|---|---|---|---|---|---|---|---|
| Dabob Bay (DB) | 1,991 | 1,929 | 1,828 | 1,728 | | 1,716 | 1,677 | 1,720 | 1,788 | 1,867 | 1,931 | 1,904 |
| *StDev* | 7 | 31 | 82 | 51 | | 36 | 51 | 38 | 31 | 52 | 22 | 41 |
| *# Obs.* | 28 | 132 | 265 | 36 | | 150 | 317 | 496 | 378 | 118 | 139 | 17 |
| Carr Inlet (CI) | 1,946 | 1,923 | 1,897 | 1,769 | 1,824 | 1,831 | 1,773 | 1,886 | 1,891 | 1,928 | 2,021 | 1,944 |
| *StDev* | 20 | 18 | 31 | 68 | 48 | 36 | 97 | 68 | 72 | 89 | 16 | 18 |
| *# Obs.* | 51 | 42 | 112 | 130 | 140 | 56 | 228 | 372 | 360 | 268 | 74 | 15 |

**Table C5**. Average monthly $f$CO$_2$ seasonal cycles for the five Washington State moorings with CO$_2$ sensors or sufficient observations to estimate $f$CO$_2$. The average monthly standard deviation (StDev) and total number of observations (# Obs.) contributing to the monthly average estimate are included.

| $f$CO$_2$ (µatm) | Jan. | Feb. | Mar. | Apr. | May | June | July | Aug. | Sept. | Oct. | Nov. | Dec. |
|---|---|---|---|---|---|---|---|---|---|---|---|---|
| Chá bă (CB) | 341 | 358 | 415 | 245 | 204 | 212 | 258 | 283 | 325 | 342 | 358 | 349 |
| *StDev* | 24 | 30 | 74 | 41 | 33 | 48 | 68 | 58 | 45 | 25 | 40 | 26 |
| *# Obs.* | 398 | 221 | 245 | 279 | 484 | 1,080 | 1,461 | 1,648 | 1,346 | 1,028 | 645 | 495 |
| Cape Elizabeth (CE) | 368 | 343 | 322 | 277 | 279 | 260 | 281 | 276 | 316 | 330 | 364 | 371 |
| *StDev* | 29 | 38 | 34 | 36 | 41 | 51 | 57 | 52 | 51 | 47 | 34 | 21 |
| *# Obs.* | 1,485 | 1,347 | 1,295 | 1,199 | 1,158 | 1,244 | 1,715 | 1,812 | 1,619 | 1,753 | 1,506 | 1,484 |
| Twanoh (TW) | 619 | 427 | 231 | 233 | 346 | 383 | 342 | 344 | 392 | 379 | 487 | 655 |
| *StDev* | 144 | 201 | 103 | 88 | 58 | 75 | 34 | 30 | 64 | 145 | 263 | 217 |
| *# Obs.* | 371 | 392 | 506 | 725 | 875 | 877 | 743 | 961 | 1,650 | 1,145 | 613 | 363 |
| Dabob Bay (DB) | 748 | 575 | 420 | 205 | | 288 | 297 | 291 | 311 | 428 | 583 | 636 |
| *StDev* | 49 | 67 | 132 | 56 | | 19 | 25 | 18 | 56 | 211 | 73 | 15 |
| *# Obs.* | 28 | 132 | 265 | 36 | | 150 | 317 | 496 | 378 | 118 | 139 | 17 |
| Carr Inlet (CI) | 731 | 679 | 602 | 261 | 351 | 377 | 264 | 450 | 420 | 502 | 843 | 659 |
| *StDev* | 64 | 67 | 137 | 133 | 84 | 84 | 167 | 177 | 184 | 241 | 69 | 57 |
| *# Obs.* | 51 | 42 | 112 | 130 | 140 | 56 | 228 | 372 | 360 | 268 | 74 | 15 |

**Table C6**. Average monthly pH seasonal cycles for the six Washington State moorings with pH sensors or sufficient observations to estimate pH. The average monthly standard deviation (StDev) and total number of observations (# Obs.) contributing to the monthly average estimate are included.

| pH | Jan. | Feb. | Mar. | Apr. | May | June | July | Aug. | Sept. | Oct. | Nov. | Dec. |
|---|---|---|---|---|---|---|---|---|---|---|---|---|
| Chá bă (CB) | 8.07 | 8.08 | 8.02 | 8.22 | 8.27 | 8.26 | 8.21 | 8.17 | 8.11 | 8.09 | 8.06 | 8.07 |
| *StDev* | 0.03 | 0.03 | 0.06 | 0.06 | 0.06 | 0.09 | 0.09 | 0.08 | 0.07 | 0.03 | 0.05 | 0.03 |
| *# Obs.* | 398 | 222 | 248 | 279 | 495 | 1,117 | 1,461 | 1,648 | 1,547 | 1,172 | 646 | 496 |
| Cape Elizabeth (CE) | 8.06 | 8.08 | 8.11 | 8.17 | 8.17 | 8.20 | 8.17 | 8.18 | 8.13 | 8.11 | 8.07 | 8.06 |
| *StDev* | 0.03 | 0.04 | 0.04 | 0.05 | 0.05 | 0.07 | 0.07 | 0.07 | 0.06 | 0.06 | 0.04 | 0.02 |
| *# Obs.* | 1,485 | 1,347 | 1,295 | 1,199 | 1,158 | 1,244 | 1,715 | 1,812 | 1,619 | 1,753 | 1,506 | 1,484 |
| Twanoh (TW) | 7.84 | 8.03 | 8.22 | 8.23 | 8.06 | 8.02 | 8.07 | 8.06 | 8.02 | 8.04 | 7.98 | 7.86 |
| *StDev* | 0.14 | 0.22 | 0.16 | 0.13 | 0.06 | 0.06 | 0.04 | 0.03 | 0.06 | 0.13 | 0.22 | 0.11 |
| *# Obs.* | 371 | 392 | 506 | 725 | 875 | 877 | 743 | 961 | 1,650 | 1,145 | 613 | 363 |
| Dabob Bay (DB) | 7.76 | 7.86 | 8.01 | 8.26 | | 8.13 | 8.11 | 8.12 | 8.11 | 8.02 | 7.87 | 7.81 |
| *StDev* | 0.03 | 0.04 | 0.14 | 0.08 | | 0.02 | 0.03 | 0.02 | 0.05 | 0.13 | 0.06 | 0.02 |
| *# Obs.* | 28 | 132 | 265 | 36 | | 150 | 317 | 496 | 378 | 118 | 139 | 17 |



| | Jan. | Feb. | Mar. | Apr. | May | June | July | Aug. | Sept. | Oct. | Nov. | Dec. |
|---|---|---|---|---|---|---|---|---|---|---|---|---|
| Carr Inlet (CI) | 7.77 | 7.80 | 7.86 | 8.21 | 8.09 | 8.04 | 8.22 | 8.00 | 8.03 | 7.98 | 7.73 | 7.81 |
| StDev | 0.03 | 0.04 | 0.09 | 0.16 | 0.10 | 0.09 | 0.20 | 0.15 | 0.16 | 0.21 | 0.03 | 0.03 |
| # Obs. | 51 | 42 | 112 | 130 | 140 | 56 | 228 | 372 | 360 | 268 | 74 | 15 |
| Bellingham Bay (BB) | 8.07 | 7.81 | 7.95 | 8.12 | 8.25 | 8.12 | 7.99 | 7.99 | 8.16 | 7.99 | 7.94 | 8.06 |
| StDev | 0.03 | 0.08 | 0.09 | 0.20 | 0.10 | 0.12 | 0.19 | 0.17 | 0.07 | 0.11 | 0.09 | 0.05 |
| # Obs. | 844 | 2,114 | 4,377 | 4,168 | 4,367 | 4,227 | 4,318 | 3,963 | 4,226 | 4,385 | 4,229 | 4,365 |


**Table C7**. Average monthly $\Omega_{Ar}$ seasonal cycles for the five Washington State moorings with
sufficient observations to estimate $\Omega_{Ar}$. The average monthly standard deviation (StDev) and total
number of observations (# Obs.) contributing to the monthly average estimate are included.

| $\Omega_{Ar}$ | Jan. | Feb. | Mar. | Apr. | May | June | July | Aug. | Sept. | Oct. | Nov. | Dec. |
|---|---|---|---|---|---|---|---|---|---|---|---|---|
| Chá bă (CB) | 1.7 | 1.6 | 1.5 | 2.4 | 2.9 | 3.0 | 2.8 | 2.7 | 2.4 | 2.2 | 2.0 | 1.9 |
| StDev | 0.2 | 0.1 | 0.2 | 0.3 | 0.3 | 0.5 | 0.5 | 0.4 | 0.3 | 0.1 | 0.2 | 0.1 |
| # Obs. | 398 | 221 | 245 | 279 | 484 | 1,080 | 1,461 | 1,648 | 1,346 | 1,028 | 645 | 495 |
| Cape Elizabeth (CE) | 1.8 | 1.8 | 1.9 | 2.1 | 2.3 | 2.6 | 2.6 | 2.7 | 2.5 | 2.3 | 2.0 | 1.8 |
| StDev | 0.2 | 0.2 | 0.2 | 0.2 | 0.3 | 0.4 | 0.4 | 0.4 | 0.3 | 0.3 | 0.2 | 0.1 |
| # Obs. | 1,485 | 1,347 | 1,295 | 1,199 | 1,158 | 1,244 | 1,715 | 1,812 | 1,619 | 1,753 | 1,506 | 1,484 |
| Twanoh (TW) | 0.9 | 1.4 | 1.9 | 2.1 | 1.6 | 1.6 | 2.0 | 2.0 | 1.7 | 1.5 | 1.3 | 1.0 |
| StDev | 0.5 | 0.7 | 0.6 | 0.5 | 0.2 | 0.2 | 0.1 | 0.1 | 0.2 | 0.5 | 0.6 | 0.3 |
| # Obs. | 371 | 392 | 506 | 725 | 875 | 877 | 743 | 961 | 1,650 | 1,145 | 613 | 363 |
| Dabob Bay (DB) | 0.8 | 1.0 | 1.3 | 2.1 | | 2.0 | 2.1 | 2.2 | 2.2 | 1.7 | 1.1 | 0.8 |
| StDev | 0.0 | 0.1 | 0.5 | 0.2 | | 0.1 | 0.1 | 0.1 | 0.2 | 0.4 | 0.1 | 0.1 |
| # Obs. | 28 | 132 | 265 | 36 | | 150 | 317 | 496 | 378 | 118 | 139 | 17 |
| Carr Inlet (CI) | 0.9 | 0.9 | 1.0 | 2.3 | 1.9 | 1.8 | 3.0 | 2.0 | 2.1 | 1.9 | 1.0 | 1.0 |
| StDev | 0.1 | 0.1 | 0.2 | 0.6 | 0.4 | 0.3 | 1.0 | 0.7 | 0.7 | 0.8 | 0.1 | 0.0 |
| # Obs. | 51 | 42 | 112 | 130 | 140 | 56 | 228 | 372 | 360 | 268 | 74 | 15 |


**Table C8**. Average monthly Revelle Factor seasonal cycles for the five Washington State
moorings with sufficient observations to estimate the Revelle Factor. The average monthly
standard deviation (StDev) and total number of observations (# Obs.) contributing to the monthly
average estimate are included.

| Revelle Factor | Jan. | Feb. | Mar. | Apr. | May | June | July | Aug. | Sept. | Oct. | Nov. | Dec. |
|---|---|---|---|---|---|---|---|---|---|---|---|---|
| Chá bă (CB) | 13.4 | 13.9 | 14.4 | 11.2 | 10.1 | 9.9 | 10.5 | 10.7 | 11.4 | 11.8 | 12.5 | 13.0 |
| StDev | 0.6 | 0.4 | 1.1 | 0.8 | 0.6 | 0.9 | 1.1 | 1.0 | 0.8 | 0.4 | 0.6 | 0.4 |
| # Obs. | 398 | 221 | 245 | 279 | 484 | 1,080 | 1,461 | 1,648 | 1,346 | 1,028 | 645 | 495 |
| Cape Elizabeth (CE) | 13.5 | 13.2 | 12.7 | 11.9 | 11.5 | 10.8 | 10.8 | 10.7 | 11.2 | 11.7 | 12.6 | 13.2 |
| StDev | 0.6 | 0.6 | 0.5 | 0.7 | 0.7 | 1.0 | 0.9 | 0.9 | 0.8 | 0.8 | 0.6 | 0.4 |
| # Obs. | 1,485 | 1,347 | 1,295 | 1,199 | 1,158 | 1,244 | 1,715 | 1,812 | 1,619 | 1,753 | 1,506 | 1,484 |
| Twanoh (TW) | 18.1 | 15.5 | 12.5 | 11.9 | 13.6 | 13.5 | 12.1 | 12.1 | 13.4 | 14.1 | 15.4 | 17.1 |
| StDev | 1.8 | 3.0 | 2.3 | 1.8 | 0.9 | 1.0 | 0.5 | 0.4 | 1.0 | 2.0 | 2.9 | 1.2 |
| # Obs. | 371 | 392 | 506 | 725 | 875 | 877 | 743 | 961 | 1,650 | 1,145 | 613 | 363 |
| Dabob Bay (DB) | 18.7 | 17.6 | 15.5 | 11.5 | | 11.9 | 11.6 | 11.2 | 11.5 | 13.5 | 16.7 | 18.4 |
| StDev | 0.2 | 0.5 | 2.1 | 1.0 | | 0.3 | 0.4 | 0.3 | 0.8 | 1.7 | 0.9 | 0.6 |
| # Obs. | 28 | 132 | 265 | 36 | | 150 | 317 | 496 | 378 | 118 | 139 | 17 |





| Carr Inlet (CI) | 18.3 | 18.0 | 17.1 | 11.6 | 13.1 | 13.1 | 10.3 | 12.8 | 12.6 | 13.6 | 17.6 | 17.7 |
|---|---|---|---|---|---|---|---|---|---|---|---|---|
| *StDev* | 0.3 | 0.5 | 1.3 | 2.3 | 1.6 | 1.5 | 2.4 | 2.2 | 2.4 | 2.9 | 0.3 | 0.3 |
| *# Obs.* | 51 | 42 | 112 | 130 | 140 | 56 | 228 | 372 | 360 | 268 | 74 | 15 |


**9. Appendix D**

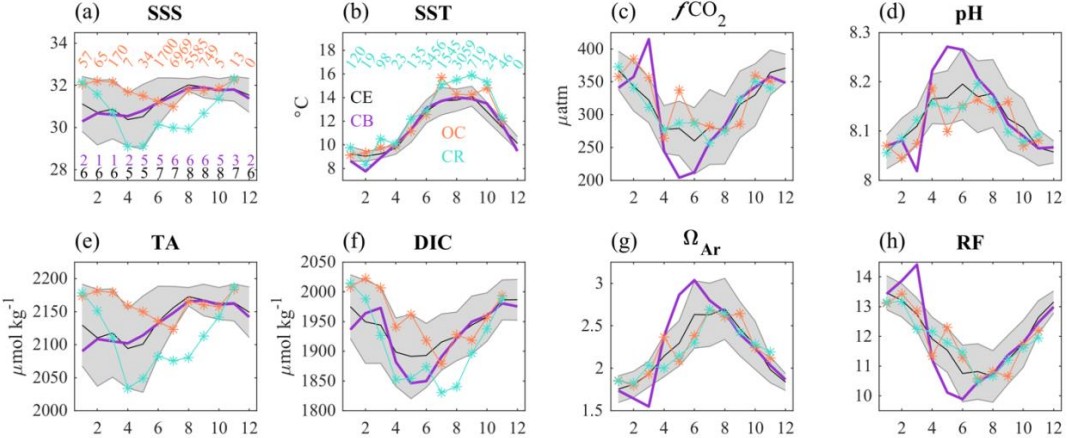

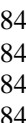


**Figure D1.** Seasonal cycles of (**a**) SSS, (**b**) SST, (**c**) $f$CO$_2$, (**d**) pH$_T$, (**e**) TA, (**f**) DIC, (**g**) $\Omega_{Ar}$, and
(**h**) RF at the Cape Elizabeth (CE) and Chá bǎ (CB) moorings, and for the Outer Coast (OC) and
Columbia River (CR) regions. Gray shading shows ±1σ for CE. Text near the bottom of **a** indicates
the number of years of moored SSS, SST, and $f$CO$_2$ observations contributing to the monthly
seasonal cycle estimates. Slanted text in **a** and **b** indicates the total number of discrete observations
contributing to the monthly seasonal cycle estimates.

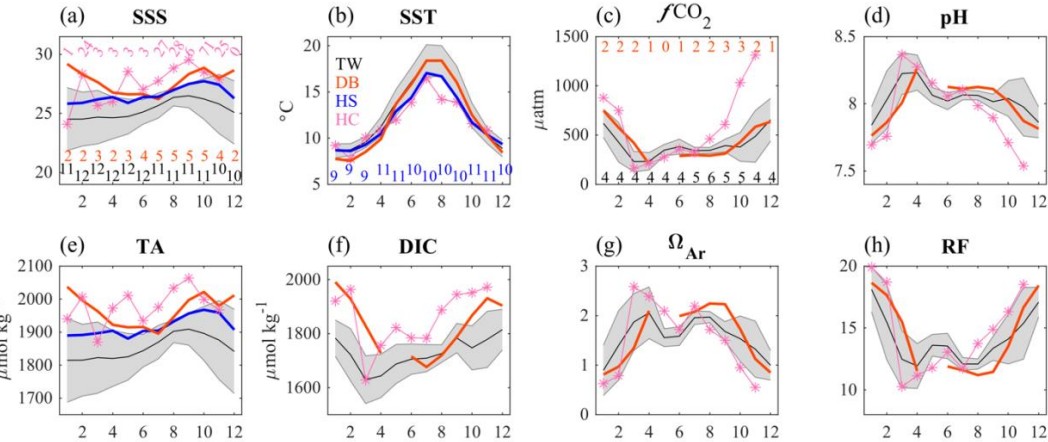

**Figure D2**. Seasonal cycles of (**a**) SSS, (**b**) SST, (**c**) $f$CO$_2$, (**d**) pH$_T$, (**e**) TA, (**f**) DIC, (**g**) $\Omega_{Ar}$, and
(**h**) RF at the Twanoh (TW), Dabob Bay (DB), and Hoodsport (HP) moorings and for the Hood



Canal (HC) region. Gray shading shows ±1σ for TW. Text near the bottom of **a** and **b** indicates
the number of years of moored SSS and SST observations contributing to the monthly seasonal
cycle estimates. Text in **c** indicates the same information but for the moored $f$CO$_2$ observations.
Text is staggered in **a** and **b** for viewing. Slanted text in **a** indicates the total number of discrete
observations contributing to the monthly seasonal cycle estimates.

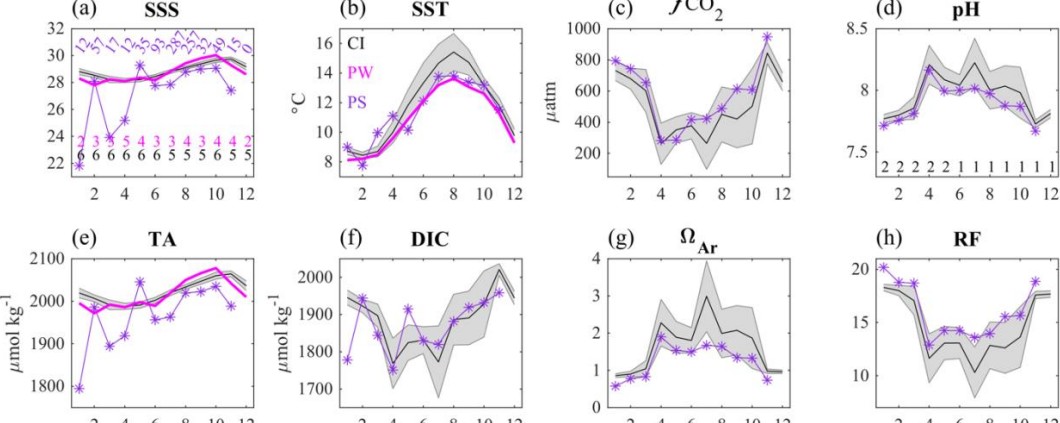


**Figure D3.** Seasonal cycles of (**a**) SSS, (**b**) SST, (**c**) $f$CO$_2$, (**d**) pH$_T$, (**e**) TA, (**f**) DIC, (**g**) $\Omega_{Ar}$, and
(**h**) RF at the Carr Inlet (CI) and Point Wells (PW) moorings and for the Puget Sound (PS) region.
Gray shading shows ±1σ for CI. Text near the bottom of **a** indicates the number of years of moored
SSS and SST observations contributing to the monthly seasonal cycle estimates. Text in **d** indicates
the same information but for the moored pH observations. Slanted text in **a** indicates the total
number of discrete observations contributing to the monthly seasonal cycle estimates.

**10. Author contributions**
A. Fassbender, S. Alin, R. Feely, and A. Sutton designed the study and contributed to the data
analysis and writing of the manuscript. J. Newton and W. Ruef provided much of the mooring and
cruise data and contributed to interpretation of the results and manuscript review. A. Devol
contributed to interpretation of the results and manuscript review. M. Keyzers wrote the sampling
plan and collected and provided data for ECY discrete samples. J. Bos, C. Krembs, and G. Pelletier
reviewed ECY data results, analyses, and the manuscript, and contributed to interpretation of the
results.

**11. Acknowledgements**
The Surface Ocean CO$_2$ Atlas (SOCAT) is an international effort, endorsed by the International
Ocean Carbon Coordination Project (IOCCP), the Surface Ocean Lower Atmosphere Study
(SOLAS) and the Integrated Marine Biogeochemistry and Ecosystem Research program
(IMBER), to deliver a uniformly quality-controlled surface ocean CO$_2$ database. The many
researchers and funding agencies responsible for the collection of data and quality control are
thanked for their contributions to SOCAT. The moored carbon observations were supported by
NOAA's Ocean Acidification Program and PMEL (NOAA FundRef 100000192). The University



of Washington acknowledges that the mooring work from Puget Sound and Chá bă assets were
supported by numerous technicians, captains/crew, and scientists, including Al Devol, John
Mickett, and Beth Curry. Cruises were often staffed by trained volunteer students through the
PRISM, NANOOS, and WOAC programs. Funding was provided by the U.S. Integrated Ocean
Observing System through NOAA to NANOOS; NOAA's Ocean Acidification Program; the State
of Washington to the Washington Ocean Acidification Center (WOAC); the Navy to the Hood
Canal Dissolved Oxygen Program; the University of Washington to PRISM and for Bellingham
Bay; and EPA. The Washington Dept. of Ecology (Christopher Krembs, Julia Bos, Mya Keyzers,
Skip Albertson, Laura Hermanson, and Carol Maloy) conducts long-term monitoring of Puget
Sound including Hood Canal, Strait of Juan de Fuca, and Washington's coastal bays, collecting
and providing quality-assured marine data since 1973. A.J.F. was supported by the Postdocs
Applying Climate Expertise (PACE) Fellowship Program, partially funded by the WOAC and
NOAA Climate Program Office and administered by the UCAR Visiting Scientist Programs. This
is PMEL contribution number 4632.

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
