# Peer review of "Seasonal Carbonate Chemistry Variability in Marine Surface Waters of the Pacific Northwest"

_Earth System Science Data, 2017_

## Referee Comment (RC1) · Anonymous Referee #1 · 7 May 2018

Peer Review

Fassbender, A.J., Alin, S.R., Feely, R.A., Sutton, A.J., Newton, J.A., Krembs, C., Bos, J., Meyzera, M., Devol, A., Ruef, W., and Pelletier, G., 2018. Seasonal carbonate chemistry variability in marine surface waters of the Pacific Northwest. Earth Syst. Sci. Discuss., doi:10.5194/essd-2017-138.

Manuscript under review for Earth System Science Data, March 2018

This manuscript uses a publicly available dataset to describe seasonal carbonate chemistry variability in surface waters in several regions through the Pacific Northwest. The dataset was created through a synthesis of both moored and discrete samples

collected from ships to explore spatial and temporal variability across monthly and 0.1 degree scales. Generally, the authors find that significant seasonal and cross-shelf (longitudinal) variability exists for each of 8 regions defined by topographic features and sea surface salinity. The authors also make a robust effort to identify and scale the errors associated with the data they describe.

In general, the authors' efforts to detail the errors associated with the compilation of their dataset represents the bulk of the analysis included here. I found this effort to be extremely meticulous and well thought through, such that the caveats associated with their analysis were clear and easy to understand. Given the careful attention given to data quality, the authors conclusions stand clearly and firmly on their own. In particular, I appreciated sections 4.2 and 4.3, which described the value of the dataset explicitly in terms of its limitations and strengths. In short, the authors were able to show that regardless of the sometimes substantial errors associated with their analysis, the qualitative interpretation of the data was unlikely to change.

For ESSD, reviewers are also asked to review the assets submitted with this manuscript. The extensive metadata record highlights the contributions of each author. The record is extensive enough to denote the unique style of each dataset, but also support the cohesive way in which the manuscript synthesized these data into a single submission. The assets themselves are also easy to understand and constructed well. For my part, I appreciated that the discrete and mooring data were separated into different .nc files for ease of use.

In summary, I am pleased to see that the authors of this manuscript have submitted a unique, useful, and complete dataset to ESSD. It should prove beneficial to others seeking to use this data as a future baseline against which to compare long-term changes or anomalous years. This large-scale environmental context will be an important asset for future research and management efforts in the Pacific Northwest. I have no major comments to contribute to the analysis given the authors attention to detail, and limited minor comments for the authors to address. I congratulate the authors on

their good work and encourage the editors to accept this manuscript for publication.

Minor Comments. This was the most carefully constructed manuscript I have reviewed in the last year. It was clear, well thought out, easy to read, and meticulously proofread. Without extensive grammar corrections, I note the following minor comments related to clarity in lines 539-548. Line 545. I encourage the authors to rephrase to exclude the word 'exceptional,' given that it can be interpreted both qualitatively (high-value, good) and quantitatively (statistical anomaly), and it is unclear which meaning the authors intend in this case.

Line 546-548. I struggle to interpret the meaning of the last sentence without additional context for the vertical scales the authors describe. I assume that the authors refer here to stratification over the upper few meters of the water column—i.e., microstratification at the extremely near surface, within the surface mixed layer—rather than to the impacts of upwelling or other vertical mixing processes occurring in areas with strong depth gradients in carbonate parameters. However, if the authors do intend to say that the magnitude of the seasonal cycle is » than upwelling, this requires some additional explanation rather than a passing reference. I encourage the authors to more specifically state their meaning or to remove this sentence.
* * *

---

## Author Comment (AC1) · 21 May 2018

We thank the Reviewer for their support of our manuscript and recognition of our efforts to clearly present the data, methods, and interpretation of results to the community.

In response to the Reviewer's minor comments we have made the following changes:

Line 544 original: "General agreement between the mooring data and relatively poorly-sampled regions is therefore somewhat exceptional, as many regional carbonate system values fall within the $\pm 1\sigma$ envelope."

Line 544 modified: "The region-based seasonal cycles fall within the $\pm 1\sigma$ envelope of

the mooring-based seasonal cycles for most carbonate system parameters, suggesting general agreement across data products."

Line 546 original: This finding underscores that seasonal variations outcompete the influence of vertical gradients associated with stratification that would work to create discrepancies between the seasonal cycle data products.

Line 546 modified: This finding underscores that seasonal variations outcompete the influence of vertical gradients in the upper water column associated with stratification, which would work to create discrepancies between the data products due to the unique depth ranges of data contributing to the mooring ($\sim$1 m) and regional ($\leq$ 10 m) seasonal cycle assessments.

———————————————

---

## Referee Comment (RC2) · D. Ianson (Referee) · 30 May 2018

Review of: Seasonal Carbonate Chemist ry Variability in Marine Surface Waters of the Pacific Northwest Fassbender et al.

General comments:

On the whole Fassbender et al. have compiled a prodigious amount of inorganic carbon data collected by different platforms in a dynamic under-sampled coastal zone. They have also made the effort to make each data-set predict the full carbonate system (using an approximation based on Salinity - to empirically determine total alkalnity.

Overall I think that this manuscript and dataset should be published and laud the authors on their hard work putting these varied data together. Where the effort could be improved I give some suggestions below.

Main points:

(1) Although this journal's aims are to produce data sets, I think that the paper would benefit from a stronger focus on the story that the data tell and mechanisms. Mechanisms are especially important to understand for readers who don't have an oceanographic background and OA is garnering concern from many interdisciplinary and political camps. The region is highly complex so responding to this comment is a tall order. (The specific comments below contain some suggestions in the region that this reviewer is familiar with, but the authors should not limit themselves to these comments.)

(2) Since the focus on combining data into one high quality data set is so important I suggest that they include data that were collected by Canadian researchers in their study region (in addition to US collection in Canadian waters), even if these data are not as readily available, something this reviewer feels badly about. (more below)

(3) Perhaps the most important general comment concerns uncertainty. The authors have considered uncertainty, but not as fully as they could. For e.g. uncertainty in discrete samples may be (is likely) underestimated, especially nearer shore - in highly stratified waters where there may be terrigineous inputs of organic matter that are assumed neglible in the open ocean assumptions. (see specific comments below) There is also considerable uncertainty in the determination of the CO2 parameters depending on which of the carbonate pairs are used,and in particular in the reaction constants. James Orr et al. have done some comprehensive work on this problem recently.

Specific comments:

This set of suggestions is up to the authors - how they handle them should not preculde
publication: As much of 20In Canada this region is not termed the 'Pacific Northwest'. Suggest: –"Pacific Northwest" —> "U.S. Pacific Northwest" – "U.S. West Coast waters" —> "U. S. West Coast and adjacent waters" or "the northern region of the California Current System".

Abstract - Suggest add more information about the results and a more concise treatment of what was done to obtain them.

Data - The authors have compiled a truly impressive amount of data.

There are more discrete and underway data available in the region and those that I know about are at:

https://www.waterproperties.ca/data/

(I anticipate that at least some would already be posted on OCADS but perhaps not at the time that the paper was prepared) This site (hosted by Fisheries and Oceans Canada) is less useable than OCADS and would contribute relatively few data relative to those contained in the full study, but still worthy of inclusion especially because they give context in the northern area of the study zone shown in Figure 1 of this submission and some of these data go back to 1973 (underway surface) and some (discrete) to 1998.

Some publications that I am aware of that contain these data include:

(a) Ianson et al. 2003 The inorganic carbon system in the coastal upwelling region west of Vancouver Island, Canada DSRI (discrete DIC and TA from the summer of 1998 (Enso!))

(b) Wong et al. 2010 Carbon dioxide in surface seawater of the eastern North Pacific Ocean (Line P), 1973–2005 DSRI (covers region in northernmost transect and Juan de Fuca Strait in this study - surface only - like SOCCATS only S and pCO2 so would require use of TA-S relationship)

(c) Ianson et al. 2016 Vulnerability of a semienclosed estuarine sea to ocean acidification in contrast with hypoxia GRL (covers the Juan de Fuca strait - discrete DIC, TA (etc) data from 2003-2012)

Lines 160-170 - Its unlikely that the true uncertainty is ts low as +/-2 umol/kg for either DIC or TA but esp. for TA and especially as one gets closer to the coast. I list one example, but it is certainly not the only one and probably not the best. E.g. Ianson et al. 2016 took >12% replicates and compared. These data show two populations in their sample area (which overlaps with this study) and they calculate an uncertainty based on replication in the field. See supp https://agupubs.onlinelibrary.wiley.com/action/downloadSupplement?doi=10.1002%2F2016GL068996attachmentId=1197 The authors don't likely have access to replication necessary, but I think that thier text should reflect the larger potential uncertainty in these open ocean methods (see above). Useful ref: Hunt et al. 2011; Contribution of non-carbonate anions to total alkalinity and overestimation of $pCO_2$ in New England and New Brunswick rivers

Section 4.1

This reader suggests that the best reference describing the circulation in the JdF is Thomson et al. 2007 JGR Estuarine versus transient flow regimes in Juan de Fuca Strait. (although certainly not nec. to support their simple mixing statement)

Perhaps more important to carbon dynamics in the tidal mixing zones (Juan de Fuca, Haro, Admiralty etc.) is the fact that at least within the former of these two Straits nutrients are never limiting to phytoplankton growth - see:
Mackas, D.L., Harrison, P.J., 1997. Nitrogenous nutrient sources and sinks in the Juan de Fuca Strait/Strait of Georgia/Puget Sound estuarine system: assessing the potential for eutrophication. Estuarine, Coastal and Shelf Science. 44, 1–21. doi:10.1006/ecss.1996.0110
and also (when its available),
Krogh et al. (2018) Risks of hypoxia and acidification in the high energy coastal envi-
ronment near Victoria, Canada's untreated municipal sewage outfalls (accepted May, 2018- Marine Pollution Bulletin)

In addition, as a function of the circulation and this tidal mixing - the seasonal cycle of surface pH is opposite in Juan de Fuca and Haro Strait when compared with the Strait of Georgia (highest pH in winter in JdF due to outer coast downwelling/upwelling circulation on the outer coast. The author's data may not show this feature but data of Ianson et al. 2016 do.

The Columbia and Fraser River appear to have different chemistry. As a result of this difference, and their unique circulation/stratification the respective impact of these rivers is radically different. The authors miss this aspect in their text. For river chemistry see https://doi.org/10.5194/bg-2017-349 - one figure has both rivers in DIC-TA space.
For Fraser river physical control of biology and as a result surface chemistry see
(a) https://agupubs.onlinelibrary.wiley.com/doi/full/10.1002/2015JC011118
(b) https://www.sciencedirect.com/science/article/pii/S027277140900002X?via

lines 425-427 - The Juan de Fuca (or Tully) eddy yields enhanced upwelling by bathymetry - a complexity that may be worth mentioning in terms of mechanisms that provide more consistent upwelling and upwelling that raises deeper isopycnals to the surface. some possible refs:
Freeland and Denman 1982: A topographically controlled upwelling center off southern Vancouver Island. JMR
Waterhouse et al. 2009 Upwelling flow dynamics in long canyons at low Rossby number. JGR
Hickey and Banas 2008: Why is the northern end of the California current system so productive? Oceanography.